# State-by-state influenza outbreaks and oversee: A Markov chain study of California and North Carolina, USA

**Asma Akter Akhi[1¤], Rabeya Akther Diba[1¤], Mohammed Abid Anwar[1¤], Tarik Mahmud Akash[1¤], Md. Kamrujjaman**[1,2¤]*

**1** Department of Mathematics, University of Dhaka, Dhaka, Bangladesh, **2** Department of Mathematics, Asian University for Women, Chattogram, Bangladesh

¤ Current Address: Department of Mathematics, University of Dhaka, Dhaka, Bangladesh.
* kamrujjaman@du.ac.bd

**Data availability statement:** All relevant data are publicly available and included as Supporting Information files with this manuscript.

## Abstract

Influenza, a significant public health concern, spreads rapidly and causes seasonal epidemics and pandemics. Mathematical models are essential tools for devising effective strategies to combat this pandemic. Various models have been utilized to study influenza's transmission dynamics and control measures. This paper presents the SEIRS (Susceptible-Exposed-Infectious-Recovered-Susceptible) model to analyze the disease's transmission dynamics. The model analyzes real data from California and North Carolina to assess trends, identify key factors, and project the nationwide spread of the disease. Subsequently, we calculate the basic reproduction number ($\mathcal{R}_0$) using the next-generation matrix method. Sensitivity analysis using Latin Hypercube Sampling (LHS) has been conducted to identify the model's most influential parameters. We graphically demonstrate how different parameters affect the exposed and infected populations, as well as the variation in the basic reproduction number with changes in parameters. We illustrate the interconnected behavior of the effective reproduction number alongside the different compartments and the basic reproduction number. We use phase plane analysis to examine the relationship between two compartments under varying parameters. Visual tools like boxplots, contour plots, and heat maps provide insights into how different factors influence the basic reproduction number and disease transmission. We investigate the stochastic behavior of the model by transforming it into a Continuous-Time Markov Chain (CTMC) model and visualizing the results graphically. We apply the SEIRS model to real influenza data, showcasing its effectiveness in analyzing transmission dynamics, predicting outbreaks, and evaluating public health strategies for better epidemic management.

## Introduction

Influenza, a highly contagious respiratory illness caused by influenza viruses, has led to major pandemics, including the 1918 (H1N1), 1957 (H2N2), and 1968 (H3N2) outbreaks [1]. Unlike seasonal flu, influenza pandemics often arise from cross-species transmission

**Funding:** The author Asma Akter Akhi acknowledged the University Grants Commission (UGC), and the University of Dhaka, Bangladesh for supplementary support of this research. No grant numbers were associated with this funding.

**Competing interests:** The authors have declared that no competing interests exist.

and viral mutations. For example, the H5N1 virus emerged in 2003 among poultry in several Asian countries, while the H1N1 pandemic in 2009 was caused by a novel virus containing a unique combination of influenza genes. The H1N1 virus, originating in Mexico and the U.S. in early 2009, rapidly spread worldwide, prompting the WHO to declare a Phase 6 pandemic on June 11, 2009 [2,3]. Influenza is a significant public health issue, infecting 5–15% of the global population annually and causing an estimated 250,000 to 500,000 deaths each year. In Bangladesh, a tropical country in the Northern Hemisphere, the seasonal influenza epidemic typically occurs during the monsoon season from May to September, mirroring patterns in the Southern Hemisphere [4,5]. The disease poses a major public health challenge due to its rapid transmission and potential for severe illness and death, especially among vulnerable groups such as the elderly or individuals with pre-existing conditions like lung disease, diabetes, cancer, or heart and kidney issues [6,7]. Influenza primarily affects the upper respiratory system—throat, bronchi, and nose—and is characterized by symptoms such as cough, chills, fever, sore throat, runny nose, muscle aches, shortness of breath, and joint pain [8].

Influenza, or the flu, comes in four types: A, B, C, and D. Humans can only contract types A and B. Among these, type A is more common and is responsible for widespread outbreaks. Type A viruses primarily originate from wild birds, while type B viruses are typically found in humans. Influenza A mutates more rapidly than Influenza B. Influenza A viruses are classified into subtypes depending on two surface proteins: hemagglutinin (H) and neuraminidase (N). There are 18 distinct hemagglutinin subtypes and 11 neuraminidase subtypes [9].

In contrast, the Influenza B virus does not have subtypes but is divided into two lineages: B/Yamagata and B/Victoria [10]. Influenza is transmitted through respiratory droplets emitted from the nose or throat, often as a result of sneezing and coughing. It can also spread when a person touches their face, nose, mouth, or eyes after coming into contact with the hands or face of someone who is infected. Breathing in these droplets can lead to contracting the flu. Because the virus changes every year, annual vaccination is recommended. Antiviral medications, such as Oseltamivir phosphate, Zanamivir, Peramivir, and Baloxavir marboxil, can treat influenza.

## Seasonal influenza trends in California and North Carolina

As for the current scenario regarding influenza disease transmission in California and North Carolina, both states are experiencing seasonal influenza activity, which typically peaks during winter. California and North Carolina are actively monitoring influenza transmission and encouraging vaccination as a key strategy to mitigate the impact of the flu this season. Public health campaigns are in place to raise awareness about the importance of flu vaccination and preventive measures to protect vulnerable populations [11]. In this article, we forecast the trend of influenza transmission in California and North Carolina through 2026, utilizing data from influenza cases detected in 2024. The insights aim to assist policymakers in making informed decisions and effective actions.

Continuous-time Markov chains (CTMC) and stochastic differential equations (SDE) have played significant roles in modeling disease transmission, with roots tracing back to the early 20th century [12]. Markov chains provide a framework for understanding transitions between states (e.g., susceptible, exposed, infected, recovered) in infectious disease dynamics. Meanwhile, the development of stochastic calculus in the mid-20th century laid the groundwork for SDEs, which incorporate random fluctuations in biological systems, allowing for a more nuanced representation of disease spread [13]. These models have been instrumental in analyzing infectious diseases, including influenza, malaria, and Covid-19, and evaluating public health interventions [14].

A key model for the spread of infectious diseases, the SIR model was first presented in the early 1900s [15,16]. Later Anderson and May added a fourth compartment (latency stage) to the SIR model. This extension is called the SEIR model [17]. Mathematical modeling of infectious diseases, such as the SEIRS (Susceptible-Exposed-Infectious-Recovered-Susceptible) model, is a crucial tool in understanding the dynamics of influenza outbreaks and informing public health interventions.

The SEIRS model categorizes the population into four compartments: susceptible ($S$), exposed ($E$), infectious ($I$), and recovered ($R$). Susceptible individuals can become exposed through contact with infectious individuals. After exposed, people go through a latent phase in which they become infected but not yet contagious. After this period, they become infectious and can transmit the virus to others. Finally, individuals recover from the infection, gaining immunity and moving to the recovered compartment. There is a high risk for the recovered to become infectious again. It can not be guaranteed that the persons who recovered from influenza and acquired immunity won't infected again.

This model is instrumental in capturing the latent period of influenza, which differentiates it from simpler models like the SIR (Susceptible-Infectious-Recovered) model. By incorporating parameters such as the transmission rate, incubation period, and recovery rate, the SEIRS model can simulate the progression of influenza within a population, predict the course of an outbreak, and evaluate the impact of interventions such as vaccination, antiviral treatments, and social distancing measures. Using data from Influenza cases in California and North Carolina between October 1, 2023, and September 24, 2024, the model assesses trends, identifies critical factors, and projects the national spread of the disease [11,18]. Fig 1 illustrates the future assumption of the disease outbreak based on real data analysis.

Considering the existing literature and historical context, this article aims to achieve the following objectives:

Develop treatment strategies to mitigate disease spread and outbreaks; conduct a theoretical analysis of the model by assessing the existence, positivity, and boundedness of solutions; analyze the basic reproduction number, and perform sensitivity analysis using Latin Hypercube Sampling (LHS) to identify the most influential parameters. Additionally, investigate the stability of equilibrium points; utilize phase plane analysis, box plots, contour plots, and heat maps to examine interactions within single and multi-compartment models; analyze the stochastic dynamics of disease transmission using the Continuous-Time Markov Chain (CTMC) model; validate the model using real-world influenza data from California and North Carolina.

The outcomes of this article related to the objectives are as follows:

The system exhibits both disease-free equilibrium (DFE) and endemic equilibrium (EE) points, and the stability of DFE and EE is achieved under specific conditions. Numerical simulations reveal that the birth and death rate ($\mu$) is the most sensitive parameter influencing disease transmission. Stochastic dynamics are consistent with the deterministic model's predictions. Moreover, contour plots, box plots, and relative influence analyses are employed to demonstrate various scenarios and their impact on the basic reproduction number ($\mathcal{R}_0$). Finally, the model's validity is supported by clinical data from influenza cases in California and North Carolina.

The main objectives and key highlights of this article are as follows:

1. Development of a comprehensive mathematical model for Influenza, incorporating classical analyses of the problem.

**Fig 1. Predicted future influenza in California.** Simulated future influenza trends based on historical data and predictive modeling.

2. Application of the model using clinical data from Influenza cases in California and North Carolina recorded between October 1, 2023, and September 24, 2024, to evaluate trends, identify key factors, and predict the national trajectory of the disease.

3. Examination of the disease-free and endemic equilibrium states, including the use of contour plots, box plots, and relative influence analyses to explore various scenarios and their impact on threshold values.

4. Determination of the most sensitive parameters influencing disease transmission using Monte Carlo Simulation methods, such as Latin Hypercube Sampling (LHS).

5. Integration of stochastic modeling alongside deterministic approaches by transforming the deterministic model into a continuous-time Markov chain (CTMC) framework.

6. Analysis of the effects of different parameters on disease transmission dynamics and the basic reproduction number threshold.

7. Exploration of strategies to control and prevent Influenza outbreaks, demonstrating that managing birth and death rates can significantly reduce disease prevalence.

This paper is organized as follows: The sections 'Formulation of Mathematical Model' and 'Positivity and Boundedness of Solution' present the construction of the mathematical model and establish the positivity and boundedness of its solutions. The fixed points, the basic reproduction number, and stability analysis are provided in the supporting materials (S1 Appendix and S2 Appendix). The section 'Factor Sensitivity Testing' performs sensitivity analysis using Monte Carlo simulations, while 'Predictive Model' presents detailed numerical analyses.

The sections 'Impact of Various Parameters' and 'Role of Transmission Rate and Recovery Rate on Basic Reproduction Number' examine the influence of key parameters ($\beta$, $\gamma$) on disease dynamics. In 'Coupled Behavior of Effective Reproduction Number and Epidemiological States', the relationships between the effective reproduction number, the various compartments, and the basic reproduction number are explored. Phase plane and box plot analyses are presented in 'Phase Plane Analysis of Two Compartments' and 'Box Plot Analysis', followed by contour and heatmap visualizations in 'Contour Plot Analysis' and 'Heatmap Analysis'. Parameter estimation via the least squares method is discussed in 'Influenza Case Study: California and North Carolina', along with inferences drawn from the computed reproduction number. Finally, 'Continuous-Time Markov Chains (CTMC)' presents CTMC simulations, 'Discussion of Computational Works' summarizes all computational findings, and 'Conclusion' highlights the key results and implications of the study.

## Formulation of mathematical model

In this paper, the SEIRS model is used to describe the dynamics of Influenza. It is a system of nonlinear differential equations that describes the rate of change of each compartment over time. There are Four-Region in this model, where, $S$, $E$, $I$, and $R$ represent numbers of susceptible, exposed, infectious, and recovered, respectively.

Let us define $\mathbb{Y} = (Y_1, Y_2, Y_3, Y_4) = (S, E, I, R)$. Then the mathematical model is given in the compact form as below,

$$\mathbb{Y}' = F(Y, t), \quad \mathbb{Y}(0) = \mathbb{Y}_0. \tag{1}$$

where, the initial conditions, $\mathbb{Y}_0 = (S_0, E_0, I_0, R_0)$, and,

$$F(Y, t) = \left[\mu N - \frac{\beta IS}{N} + \omega R - \mu S, \frac{\beta IS}{N} - \sigma E - \mu E, \sigma E - \gamma I - \mu I, \gamma I - \omega R - \mu R\right].$$

Note that, the expanded model is available in the supplementary file (S1 Appendix) for readers. The total population is,

$$N(t) = S(t) + E(t) + I(t) + R(t).$$

with $N_0 = S_0 + E_0 + I_0 + R_0$, and $t \in (0, \infty)$.

There is a latent period between the susceptible and infectious (exposed) groups. At a rate of $\beta SI/N$, the individuals transition from the susceptible class to the exposed class, where they stay for an average of $1/\sigma$ before entering the $I$ group. So after a certain period of $1/\omega$, the recovered individuals again fall into the $S$ group. Background deaths from other causes will occur at a rate of $\mu$ for all categories. Otherwise, background deaths are balanced by births into $S$ at a rate $\mu N$ [19]. The total population size $N$ satisfies,

$$\frac{dN}{dt} = \frac{dS}{dt} + \frac{dE}{dt} + \frac{dI}{dt} + \frac{dR}{dt} = \mu(N - S - E - I - R) = \mu[N - (S + E + I + R)]$$

$$= \mu[N - N] = 0.$$

Descriptions of the parameters of the model and their units are given in Table 1. The schematic diagram of this compartmental model (1) is shown in Fig 2.

**Table 1. Description of the parameters.**

| Parameter | Description | Parameter Values | Source |
|---|---|---|---|
| $\beta$ | Transmission rate | $0.21\ day^{-1}$ | [19] |
| $\sigma$ | Latency rate | $\frac{1}{7}\ day^{-1}$ | [19] |
| $\gamma$ | Recovery rate | $\frac{1}{14}\ day^{-1}$ | [19] |
| $\omega$ | Loss of immunity rate | $1\ year^{-1}$ | [19] |
| $\mu$ | Birth and death rate | $\frac{1}{76}\ year^{-1}$ | [19] |

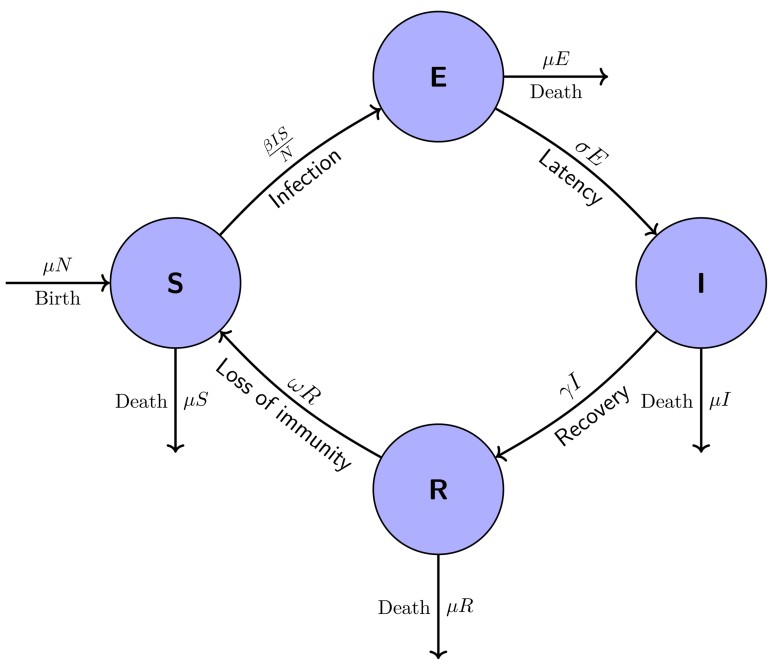

**Fig 2. Diagrammatic representation of the SEIRS model's transmission dynamics.**

Following preliminary simulations, the model's basic reproduction number is,

$$\mathcal{R}_0 = \frac{\sigma\beta}{(\sigma + \mu)(\gamma + \mu)}.$$

The disease-free equilibrium (DFE) point of the model is $P_0 = (N, 0, 0, 0)$, and the required endemic equilibrium (EE) becomes,

$$P_1 = \left( \frac{N}{\mathcal{R}_0}, \frac{(\mathcal{R}_0 - 1)\,\mu N(\gamma + \mu)(\omega + \mu)}{\sigma\,[\beta(\omega + \mu) - \omega\gamma\mathcal{R}_0]}, \frac{(\mathcal{R}_0 - 1)\,\mu N(\omega + \mu)}{[\beta(\omega + \mu) - \omega\gamma\mathcal{R}_0]}, \frac{(\mathcal{R}_0 - 1)\,\gamma\mu N}{[\beta(\omega + \mu) - \omega\gamma\mathcal{R}_0]} \right).$$

## Positivity and boundedness of solutions

The positivity and boundedness of the solutions have monumental aspects in the analysis of an epidemiological model [20]. To check the positivity and boundedness of the given SEIRS

model, we need to ensure that the solutions of the differential equations remain non-negative and do not grow without bound over time. Biologically, positivity implies the population will survive a long time [21]. To ensure positivity, we need to show that $S(t)$, $E(t)$, $I(t)$, and $R(t)$ remain non-negative for all $t \geq 0$, given non-negative initial conditions $S_0 \geq 0$, $E_0 \geq 0$, $I_0 \geq 0$, and $R_0 \geq 0$.

To ensure boundedness, we need to show that the total population, $N(t) = S(t) + E(t) + I(t) + R(t)$ remains bounded through out the time.

The system (1) is locally Lipchitz in $\mathbb{Y}$ and maintains existence and uniqueness theory for the differential equation and implies that a solution exists in $(0,T)$ for some $T < \infty$. Now, to prove positivity and boundness, we will show that the set $\zeta$ is invariant under the system flow.

$$\zeta = \{\mathbb{Y} \in \mathbb{R}^4 : Y_i \geq 0, \sum_{i=1}^{4} Y_i \leq N\}.$$

The solution occurs globally in time, as indicated by $T = \infty$, as the set $\zeta$ is positively invariant under the flow $Y(t)$ for $t \in (0, T)$.

**Theorem 1.** *The closed region $\zeta = \{\mathbb{Y} \in \mathbb{R}^4 : Y_i \geq 0, \sum_{i=1}^{4} Y_i \leq N\}$ is positive and bounded for the system (1).*

*Proof*: The boundary component $\kappa$ for $i = \overline{1,5}$, are denoted by,

$$\kappa_i = \{\mathbb{Y} \in \zeta : Y_i = 0, \quad i = \overline{1,4}\},$$

$$\kappa_5 = \{\mathbb{Y} \in \zeta : \sum_{i=1}^{5} Y_i = K\},$$

where, $\partial\zeta = \cup_{i=1}^{5} \kappa_i$. Furthermore, the inward normal $\mathbf{n}_i$, defined for the boundary segments $\kappa_i, i = \overline{1,4}$ is, $\mathbf{n}_i = \pi_i = (0, \cdots, 1, \cdots, 0)$ where the i-component is nonzero, and $\mathbf{n}_5 = (-1, -1, \cdots, -1, -1)$. The positive linear combination of the boundary segments' inward normal is denoted by $\mathbf{n}$. We will do our proof if, $\mathbf{n} \cdot \mathbb{Y}(t) \geq 0$. Now, for $i = \overline{1,4}$ and $\sum_{i=1}^{5} Y_i = N$,

$$\pi_1 \cdot \mathbb{Y}' = \mu N \geq 0, \text{ for } \mathbb{Y} \in \kappa_1,$$

$$\pi_2 \cdot \mathbb{Y}' = \frac{\beta Y_1 Y_3}{N} \geq 0, \text{ for } \mathbb{Y} \in \kappa_2,$$

$$\pi_3 \cdot \mathbb{Y}' = \sigma Y_2 \geq 0, \text{ for } \mathbb{Y} \in \kappa_3, \quad \text{and}$$

$$\pi_4 \cdot \mathbb{Y}' = \gamma Y_3 \geq 0, \text{ for } \mathbb{Y} \in \kappa_4.$$

Considering, $\sum_{i=1}^{5} Y_i = N$ we get,

$$\frac{dN}{dt} = \frac{dS}{dt} + \frac{dE}{dt} + \frac{dI}{dt} + \frac{dR}{dt} = \mu N - \frac{\beta IS}{N} - \mu S + \omega R + \frac{\beta IS}{N} - (\mu + \sigma)E + \sigma E - (\mu + \gamma)I + \gamma I - (\omega + \mu)R.$$

Simplifying, we get,

$$\frac{dN}{dt} = \frac{dS}{dt} + \frac{dE}{dt} + \frac{dI}{dt} + \frac{dR}{dt} = \mu N - \mu(S + E + I + R) = \mu N - \mu N = 0.$$

Thus,

$$\frac{dN}{dt} = 0. \qquad (2)$$

Let $N = N_0$ for a constant $N_0$. On $\kappa_5$, we have,

$$\mathbf{n}_5 \cdot \mathbb{Y}' = 0 \geq 0.$$

Integrating (2) with respect to t, we obtain,

$$N = N_0.$$

which is constant and hence bounded.

which completes the proof, since $\zeta$ is positively invariant. It should be mentioned that the solution $Y_t \in \zeta$ exists for the selected initial conditions $Y_0 \in \zeta$. □

## Factor sensitivity testing

Modeling infectious disease dynamics, such as influenza, requires careful consideration of how uncertain factors impact predictions. In the SEIRS model for influenza, several key parameters need to be analyzed for sensitivity using Monte Carlo simulation. These parameters include the contact rate ($\beta$), latency rate ($\sigma$), recovery rate ($\gamma$), loss of immunity rate ($\omega$), and the birth/death rate ($\mu$). To account for variability in these parameters, we assign probability distributions based on their expected behavior. The maximum and lowest values for each of the seven LHS parameters, as detailed in Table 2, are established according to the designated model. Notably, the baseline value for each LHS parameter is established at the midpoint between its minimum and maximum thresholds. The table presents the partial rank correlation coefficient (PRCC) values and corresponding p-values for the parameters $\beta$, $\sigma$, $\gamma$, $\omega$, and $\mu$ across the four compartments in an SEIR model: susceptible ($S$), exposed ($E$), infected ($I$), and recovered ($R$). The PRCC values indicate the sensitivity of each parameter on the respective compartment, while the p-values provide statistical significance, with smaller p-values indicating stronger confidence in the results. Figs 3 and 4 illustrate the PRCC and corresponding p-values, with detailed numerical results provided in Table 3.

The sensitivity analysis of the SEIRS model identifies key parameters that significantly influence disease dynamics. Among them, the death rate ($\mu$) is the most sensitive, particularly affecting the susceptible ($S$) and recovered ($R$) compartments. Higher mortality reduces

**Table 2. Minimum, maximum, and baseline values of parameters.**

| Parameter | Minimum | Baseline | Maximum |
|---|---|---|---|
| $\beta$ | 0.1 | 0.2 | 0.3 |
| $\sigma$ | 0.05 | 0.1 | 0.2 |
| $\gamma$ | 0.07 | 0.1 | 0.15 |
| $\omega$ | 0.01 | 0.05 | 0.1 |
| $\mu$ | 0.02 | 0.04 | 0.08 |

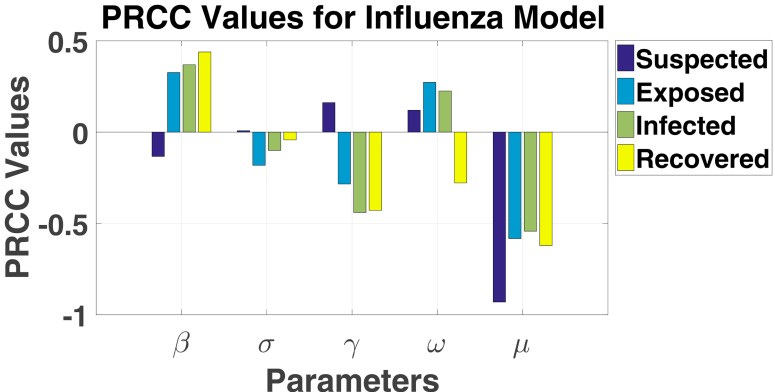

**Fig 3. PRCC values for influenza model.** PRCC values for SEIRS influenza model parameters.

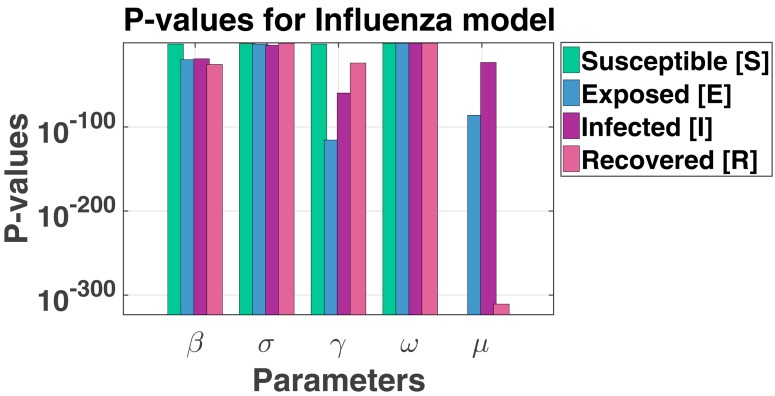

**Fig 4. P-values for influenza model.** p-values for SEIRS influenza model parameters.

**Table 3. PRCC and p-values for parameters across SEIRS model compartments.**

| Parameter | Susceptible ($S$) | | Exposed ($E$) | | Infected ($I$) | | Recovered ($R$) | |
|---|---|---|---|---|---|---|---|---|
| | PRCC | p-value | PRCC | p-value | PRCC | p-value | PRCC | p-value |
| $\beta$ | -0.065394 | 0.038682 | 0.288193 | 1.403556e-20 | 0.281844 | 1.023847e-19 | 0.327418 | 2.038824e-26 |
| $\sigma$ | -0.053009 | 0.093860 | -0.067179 | 3.365826e-02 | -0.099829 | 1.573287e-03 | 0.032507 | 3.044510e-01 |
| $\gamma$ | 0.066264 | 0.036160 | -0.639933 | 2.684737e-116 | -0.485972 | 2.138235e-60 | -0.316267 | 1.146825e-24 |
| $\omega$ | -0.038164 | 0.227898 | -0.033694 | 2.871176e-01 | -0.032302 | 3.075036e-01 | -0.030704 | 3.320606e-01 |
| $\mu$ | -0.987521 | 0.000000 | -0.568910 | 7.930854e-87 | -0.312343 | 4.548744e-24 | -0.871661 | 2.365631e-311 |

the susceptible population by removing individuals before they progress through disease states and limits the Recovered population by decreasing survival rates. The recovery rate ($\gamma$) strongly impacts the exposed ($E$) and infected ($I$) compartments. An increase in $\gamma$ accelerates the transition from infection to recovery, reducing the number of infectious individuals and shortening the duration of infection, thereby curbing disease transmission.

Similarly, the transmission rate ($\beta$) plays a critical role in the exposed ($E$) and infected ($I$) compartments. Higher $\beta$ increases the flow of individuals from susceptible to exposed, driving infection spread. Its influence on the recovered compartment reflects the eventual

recovery of those who contract the disease, linking transmission directly to recovery dynamics. In contrast, the progression rate ($\sigma$) and loss of immunity rate ($\omega$) exhibit minimal sensitivity across all compartments. Changes in $\sigma$, which governs the transition from exposed to infected, have a negligible effect on disease progression, while variations in $\omega$ have little impact on overall population dynamics, indicating limited influence of immunity waning in this model.

The death rate ($\mu$), recovery rate ($\gamma$), and transmission rate ($\beta$) are the most influential parameters, shaping the flow of individuals through the susceptible, exposed, infected, and recovered compartments. These parameters are critical for understanding and controlling disease spread, making them primary targets for public health interventions. Conversely, the progression rate ($\sigma$) and loss of immunity rate ($\omega$) are less impactful, suggesting a lower priority in influencing population-level outcomes. Further discussion on PRCC, LHS, and p-values for all relevant parameters can be found in the supporting file (S2 Appendix).

## Predictive model

### Chronological observation of the model

The dynamics of the compartmental model (1) are analyzed and visualized in Figs 5 and 6.

From Fig 5, we see that most of the population is susceptible to the disease at the initial time. As time progresses, some people get infected with the disease and some fall into the recovered class by gaining immunity. In the beginning, the infected individuals' number will grow. Also, the number of recovered individuals increases but the number of susceptible individuals decreases with the increase of infected individuals. After a certain time, the number of infected individuals gradually decreases and slows down the growth rate of the disease. But recovered individuals will gradually increase. After reaching its peak, the epidemic curve

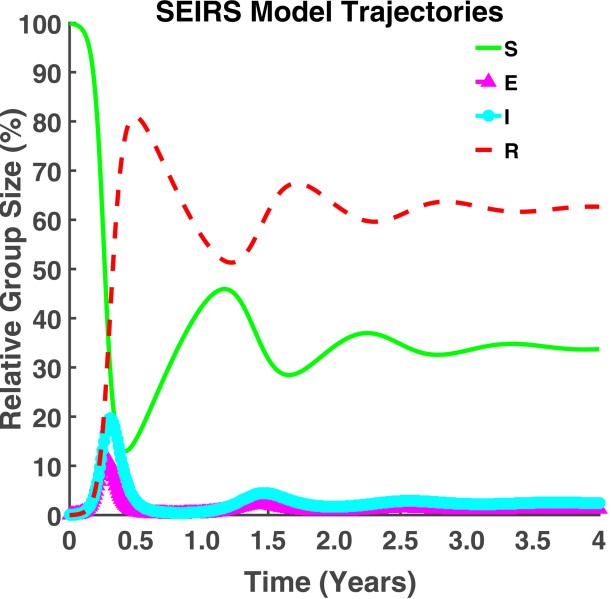

**Fig 5. SEIRS model trajectories.** Evolution of susceptible (*S*), exposed (*E*), infected (*I*), and recovered (*R*) populations over time.

**Trajectories of Exposed and Infected Population**

Fig 6. Trajectories of exposed and infected population. Dynamical behavior of exposed ($E$) and infected ($I$) populations over time where $(S_0, E_0, I_0, R_0) = (99.9, 0.1, 0, 0)$ [19].

will start to decline. Figs 5 and 6 indicate that the epidemic curve is oscillating. The oscillation in both figures represents multiple waves of infections and recovery over time. Fig 6 indicates that after a long time, the exposed and infected individuals became parallel and never approached zero. Additionally, we observe that as the number of exposed people rises, so does the number of infected people. At this time, the recovered class surpasses the susceptible class.

## Impact of various parameters

In model (1) different types of parameters have been introduced such as $\beta$, $\gamma$, $\omega$, $\sigma$, and $\mu$ where $\beta$ denotes transmission rate and $\gamma$ represents recovery rate. Depending on the values of these parameters all classes change. For instance, the number of susceptible populations grows more quickly than before if the rate of transmission rises. How much these parameters affect the total number of susceptible populations, exposed populations, and infected populations are described below.

We examine from Fig 7 that the total number of exposed population ($E(t)$) has increased with the increase in transmission rate ($\beta$) and vice versa. For $\beta = 76.65$, approximately 12% of the total population will be exposed. For a 50% increase in transmission rate from the exact value, approximately 18.02% of the total population will be exposed and for the case of a 50% decrease, it will be approximately 2.18% of the total population.

From Fig 8, it is evident that as $\beta$ increases, so does the infected population ($I(t)$), similar to the exposed population. From the figure, we can also say that if the transmission rate increases by 50%, a maximum of 28.37% of the total population will be infected and for the case of 50% decrease in $\beta$ from the exact rate ($\beta = 76.65$), it will be approximately 4.3%.

From both Figs 7 and 8, it is also evident that the highest number of the population will be exposed and infected in between the first six months. After reaching some higher peak point (the point where the highest number of susceptible are infected), the transmission rate will decrease gradually and will increase again. From these oscillating graphs, we can say that the

**Impact of Transmission Rate ($\beta$) on Exposed Population**

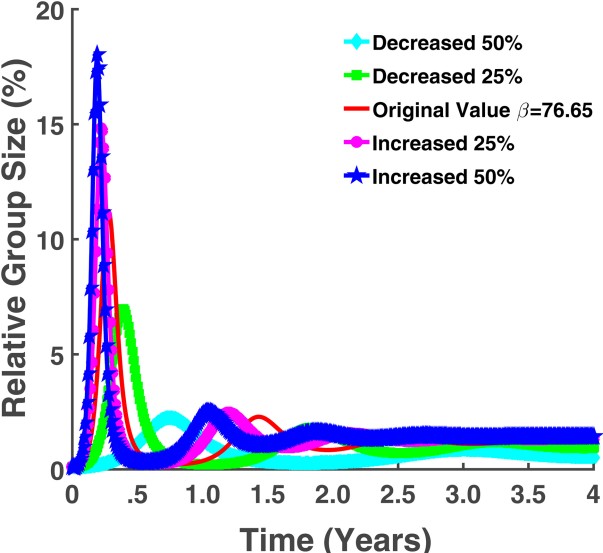

**Fig 7. Impact of transmission rate ($\beta$) on exposed population.** Effect of transmission rate ($\beta$) on exposed population ($E(t)$).

**Impact of Transmission Rate ($\beta$) on Infected Population**

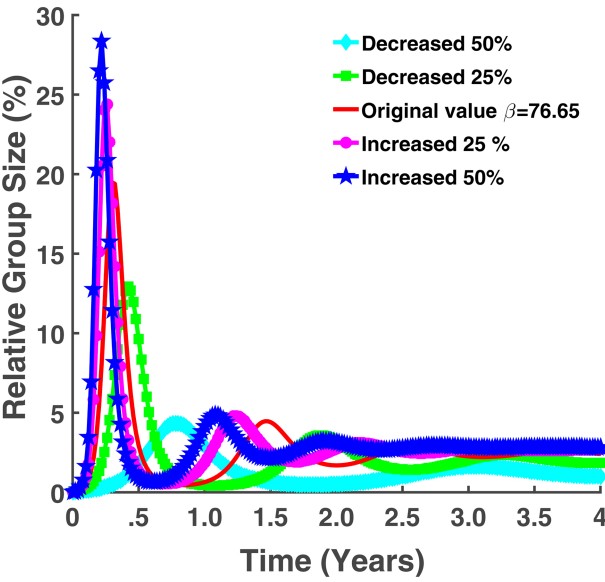

**Fig 8. Impact of transmission rate ($\beta$) on infected population.** Effect of transmission rate ($\beta$) on infected individuals ($I(t)$).

graph is fluctuating but the transmission rate will never become zero. Hence the number of exposed and infected will never become zero throughout the time. In other words, the illness is not going away. The populace will continue to experience it.

From Fig 9 effectively demonstrates how varying $\beta$ influences the dynamics of the susceptible population over time. we can see that the number of susceptible become infected as the transmission rate ($\beta$) increases and vice versa. This indicates that more people have a chance of being infected. In this case, most of the population falls into the exposed class or into the infected class (almost all of them are infected). The disease is spreading more rapidly due to the increase in transmission rate. When $\beta$ is reduced by 50%, a slower initial decline is observed in the susceptible population, with the percentage remaining relatively higher over time. When the transmission rate is increased by 25%, a more rapid initial decline is shown compared to the original value. For increased $\beta$ values (25% and 50%), the population drops quickly and stabilizes at a low level. For decreased $\beta$ values (25% and 50%), the population experiences a slower decrease and settles at a relatively higher percentage. A lower $\beta$ value results in a slower decrease, with the susceptible population stabilizing at a higher percentage.

From both Figs 10 and 11, it is noticeable that when $\gamma$ falls, where $\gamma$ is the individuals' recovery rate, the number of exposed and infected persons rises. An increase in the recovery rate implies that the infected individuals are recovering from the infection at a faster rate. For $\gamma = 365/14$, approximately 14% of the total population remains in the incubation period where approximately 19% of the total population becomes infected. If the recovery rate decreases by 25%, we can notice from Fig 10 that about 13.63% of the total population has the chance of becoming infected which is approximately 8.9% for the case of 25% increase in $\gamma$. Fig 11 also indicates that approximately 13.26% of the total population becomes infected in this case. The number of infected individuals may increase for a certain period. Following that, the number of infected people will progressively decline until rising once more. There will be no such cases where the infection rate will be zero. Hence to keep the disease under control, we have to keep the recovery rate as large as possible. Fig 12 indicates that a higher recovery rate ($\gamma$) results in a faster initial drop in the susceptible population but stabilizes at higher levels. Lower $\gamma$ values lead to a more gradual decline, stabilizing at lower levels.

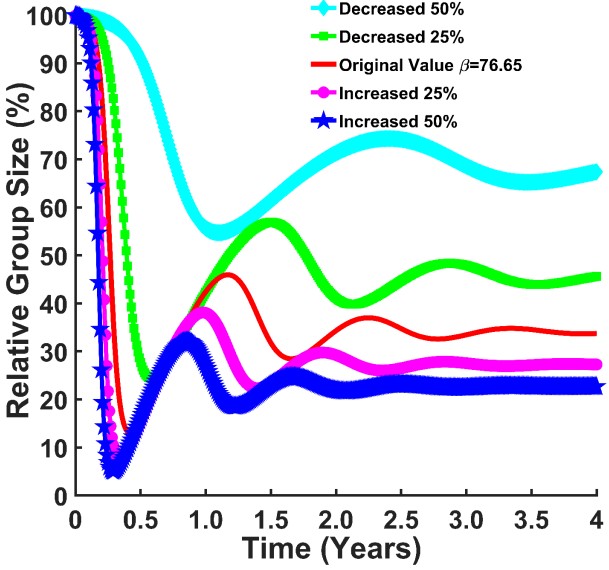

**Fig 9. Impact of transmission rate ($\beta$) on susceptible population.** Effect of transmission rate ($\beta$) on susceptible population ($S(t)$).

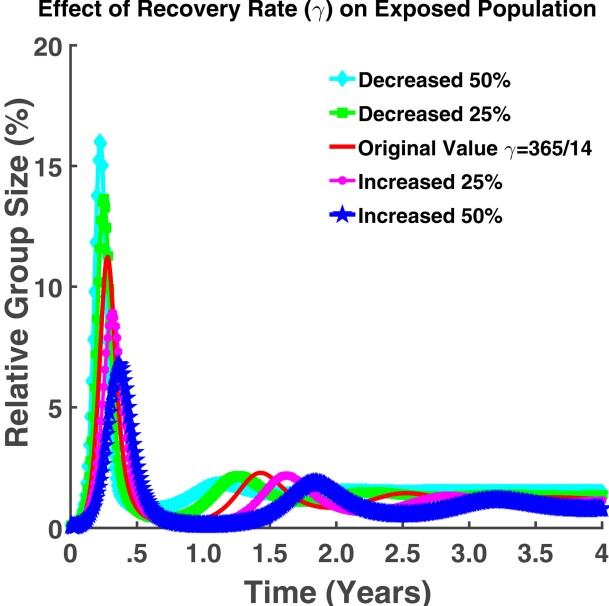

**Fig 10. Effect of recovery rate (γ) on exposed population.** Effect of recovery rate (γ) on exposed population (*E*).

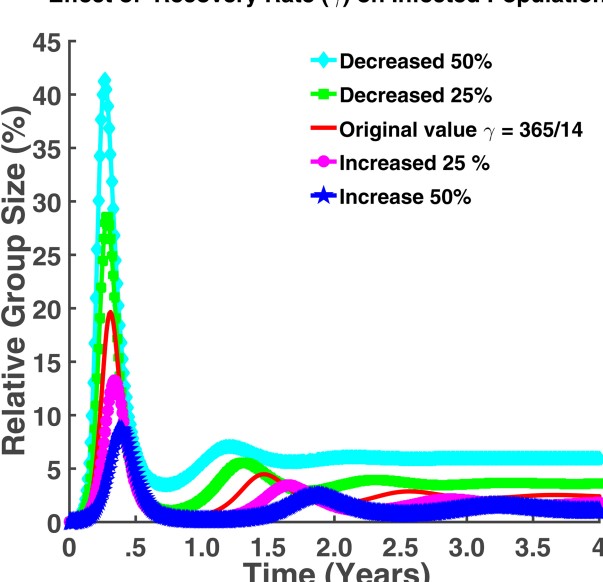

**Fig 11. Effect of recovery rate (γ) on infected population.** Effect of recovery rate (γ) on infected population (*I*).

In both cases, we can see that the susceptible population size becomes constant after a certain period. The number of susceptible is highest whereas the exposed and infected are the lowest. If the rate at which people come into contact (transmission rate) with an infection exceeds the recovery rate, the number of infected individuals might keep growing indefinitely, leading to a continuous or increasing number of cases without ever peaking.

**Fig 12. Effect of recovery rate ($\gamma$) on susceptible population.** Effect of recovery rate ($\gamma$) on susceptible population ($S$).

**Role of transmission rate and recovery rate on basic reproduction number.** When making decisions during an epidemic, the basic reproduction ratio ($\mathcal{R}_0$) is crucial. We have demonstrated how this number varies for various values of $\beta$ and $\gamma$ using the model (1). Fig 13 indicates that the basic reproduction number increases as the transmission rate ($\beta$) increases and vice versa. Fig 14 indicates that $\mathcal{R}_0$ decreases as the recovery rate ($\gamma$) increases. For a large value of $\gamma$, it will be close to zero. The basic reproduction number ($\mathcal{R}_0$) will be very large when $\gamma$ is close to zero. The infection rate will be highest at that point.

**Coupled behavior of effective reproduction number and epidemiological states.** This section presents computational results for both the basic reproduction number ($\mathcal{R}_0$) and effective reproduction number ($\mathcal{R}_e$) within a compartmental epidemiological framework, integrating theoretical derivations with computational validation. We characterize disease transmission dynamics across distinct compartments and quantify how population structure and intervention effects modulate the transition from $\mathcal{R}_0$ to $\mathcal{R}_e$.

Figs 15, 16, 17, and 18 illustrate the time evolution of population compartments and the effective reproduction number, $\mathcal{R}_e = \mathcal{R}_0 * \dfrac{S(t)}{N}$ in the SEIRS model over 5 years. Fig 15 presents a comprehensive view of all key compartments: susceptible ($S$), exposed ($E$), infectious ($I$), and recovered ($R$) along with the effective reproduction number ($\mathcal{R}_e$), capturing the complete disease transmission and recovery cycle. Fig 16 focuses on the interaction between $S$, $R$, and $\mathcal{R}_e$ to emphasize how the dynamics of the population influences the effective spread of infection over time. Fig 17 highlights the inverse relationship between the susceptible population and $\mathcal{R}_e$, indicating that a decrease in susceptibility leads to reduced transmission potential. In contrast, Fig 18 shows the correlation between the recovered population and $\mathcal{R}_e$, reflecting how increasing immunity within the population contributes to the long-term decline in disease spread.

**Basic Reproduction Number vs Transmission Rate**

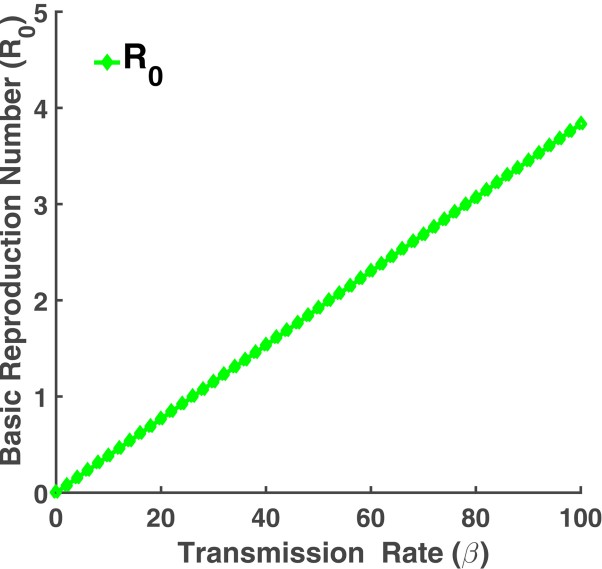

**Fig 13. Basic reproduction number vs transmission rate.** Variation of the basic reproduction number ($\mathcal{R}_0$) with transmission rate ($\beta$).

**Basic Reproduction Number vs Recovery Rate**

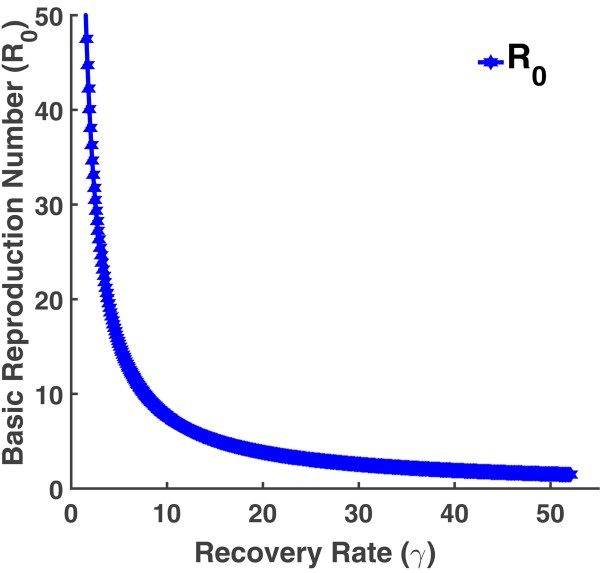

**Fig 14. Basic reproduction number vs transmission rate.** Variation of the basic reproduction number ($\mathcal{R}_0$) with recovery rate ($\gamma$).

Fig 19 illustrates how the effective reproduction number ($\mathcal{R}_e$) varies with changes in the susceptible population ($S$) and the basic reproduction number ($\mathcal{R}_0$). As expected from the relation $\mathcal{R}_e = \mathcal{R}_0 * \dfrac{S(t)}{N}$, $\mathcal{R}_e$ increases with both higher values of $\mathcal{R}_0$ and a larger susceptible population. The gradient shift from blue to red indicates that when most of the population

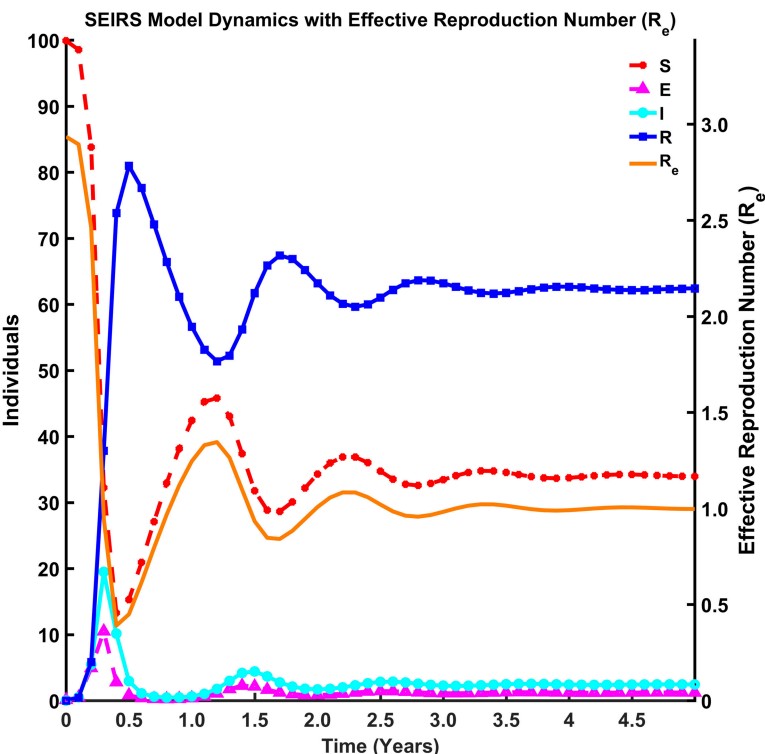

**Fig 15. SEIRS model dynamics with effective reproduction number $R_e$.** Dynamics of all compartments: *S*, *E*, *I*, *R*, and the effective reproduction number ($\mathcal{R}_e$).

is susceptible and $\mathcal{R}_0$ is high, $\mathcal{R}_e$ exceeds the threshold of 1, considering potential outbreak conditions. Conversely, lower *S* and $\mathcal{R}_0$ values result in $\mathcal{R}_e < 1$, suggesting disease decline.

Thus, Figs 15-18 depict the relationship between each compartment of the SEIRS model and the effective reproduction number. At the initial stage of an outbreak, the effective reproduction number and the basic reproduction number are the same, as the entire population is susceptible. The value of this number is approximately 3, implying that each infected individual is expected to infect about 3 others at the beginning of an outbreak. As time advances, $\mathcal{R}_e$ adjusts according to the proportion of the population that remains susceptible, using the formula $\mathcal{R}_0 * \dfrac{S(t)}{N}$ where *N* is the total population. This figure provides a realistic view of how an outbreak unfolds over time. As *I(t)* (the number of infected individuals) declines and *R(t)* (the number of recovered individuals) rises, the effective reproduction number ($\mathcal{R}_e$) drops over time due to fewer susceptibles and a faster recovery rate. Oscillations suggest a pattern of recurring epidemics, most likely brought on by declining immunity. $\mathcal{R}_e$ falls below 1, signifying epidemic control, as immunity increases (via recovery).

## Phase plane analysis of two compartments

An effective way to understand the epidemic wave is phase plane analysis. Figs 20, 21, and 22 show the trajectories of *S*, *E*, and *R* as a function of *I* while the transmission rate varies. Figs 24, 25, and 26 represent the trajectories of *S*, *E*, and *R* as a function of *I* while the recovery rate varies. Figs 20-23, and 27 present the trajectories of *R* as a function of *S*. All of them are spiral.

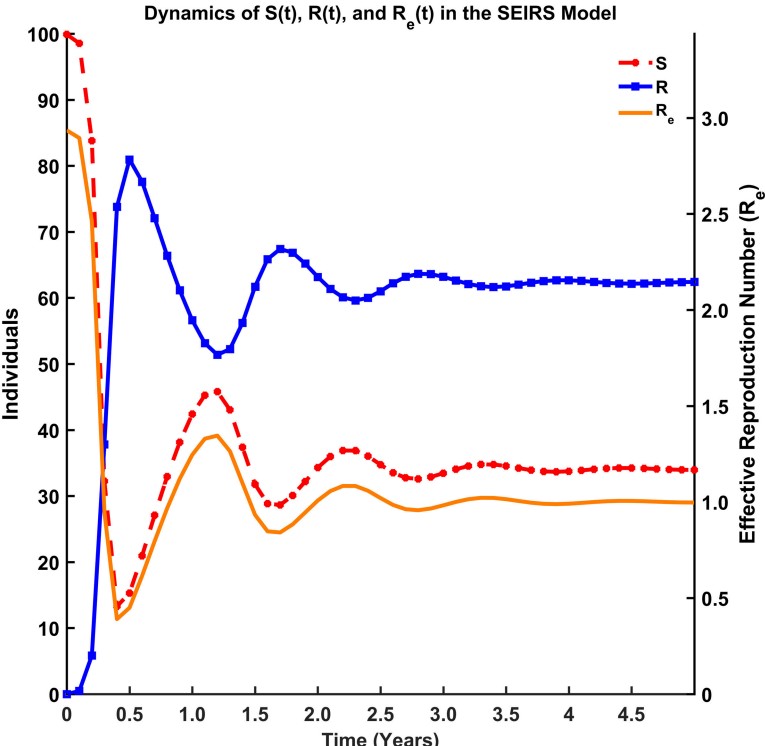

**Fig 16. Dynamics of $S(t)$, $R(t)$, and $R_e(t)$ in the SEIRS model.** Dynamics of $S$, $R$, and $\mathcal{R}_e$ highlighting long-term interactions.

From Fig 20, we see that at the initial time point, all individuals are considered as susceptible since no one has become infected yet. Hence the number of infected individuals is zero. With the progress of time, the susceptible population decreases and the number of infected individuals increases. Then the graph moves towards the infected axis. Most of the people become infected when the transmission rate is highest. Fig 21 indicates that the infected individuals grow faster with the faster growth of the exposed individuals. It affects the overall size of the infected population because more people transit into the infected class from the exposed class. Hence the disease spreads rapidly. Thus, to keep the disease under control, we have to minimize the exposed class so that the transmission rate becomes as low as possible. This relationship plays a crucial role to control the disease and take necessary steps.

Fig 22 shows the relationship between infected individuals and recovered individuals. After spending some period as infectious, infected individuals recover, and then they fall into the recovered class. Thus the number of active infected individuals reduces. It is not possible to get 100% of the total population recovered. The term 'no infected individual' does not imply that all people have recovered from the disease. It implies that the disease exists in nature either they will be considered as the susceptible population or as the exposed population (the virus is present in the individual's body but in inactive condition). From Fig 23, we see that we can have at most 80%-90% of the total population as the recovered population. After recovering, the individuals get immunity. But this immunity is not permanent. Immunized people may lose their immunity and fall into the susceptible class after a certain time. At the initial stage, all people are considered as the susceptible population. In this case, the

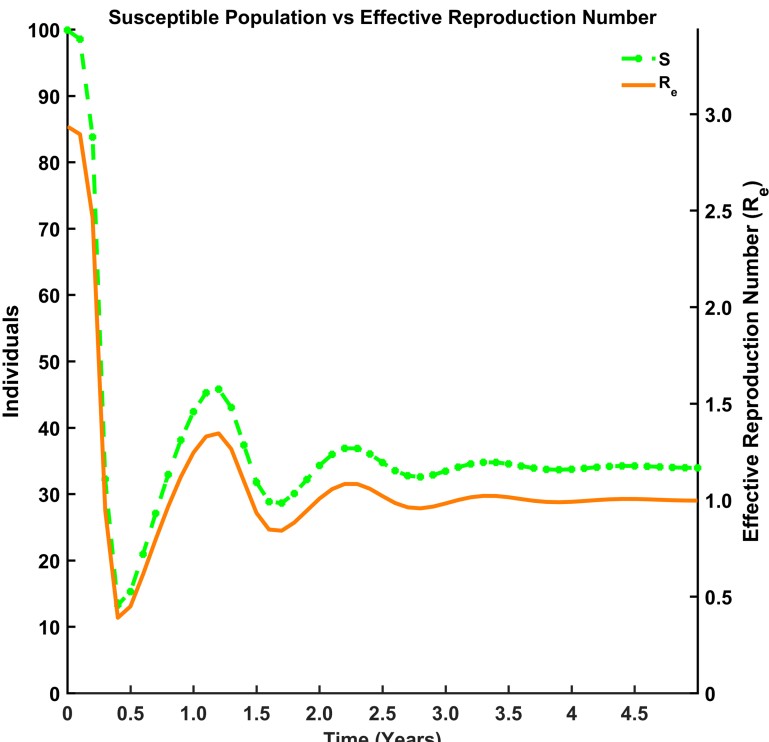

**Fig 17. Susceptible population vs effective reproduction number.** Comparison between the susceptible population
(S) and $\mathcal{R}_e$.

exposed, infected and recovered population becomes zero. The recovered population is the
highest when the susceptible population is the lowest in number.

Fig 24 depicts the dynamics of the system where susceptible individuals and infected individuals vary over time. The parameter $\gamma$ also varies which is considered as the recovery rate.
When the susceptible population increases and the infected population decreases, all of these
trajectories move toward the susceptible axis. But when the number of infected individuals increases, these trajectories move toward the infected axis. We also see that susceptible
individuals shift to infected individuals rapidly if the recovery rate decreases. This behavior
indicates that the epidemic is spreading briskly.

Fig 25 indicates the dynamics of the system while the exposed population and susceptible population vary and others remain constant. These trajectories allow us to visualize the
dynamics of the epidemic by plotting the exposed population on one axis and the infected
population on another axis. It helps us to understand how fast this epidemic is spreading
when the recovery rate is very low or very high from the actual rate. When the value of the
infected population increases, these trajectories shift towards the infected axis. At first, both
the exposed population and the infected population increase. After spending some latent
period, the virus starts to show some symptoms in the affected body and then falls into the
infected class.

From Fig 26, we see how this system behaves when the recovered population and the
infected population vary and others remain constant. People heal faster when the recovery
rate is high. More people shift to the recovered class from the infected class. The graph moves
towards the recovered axis. Fig 27 shows the number of recovered is highest when the number

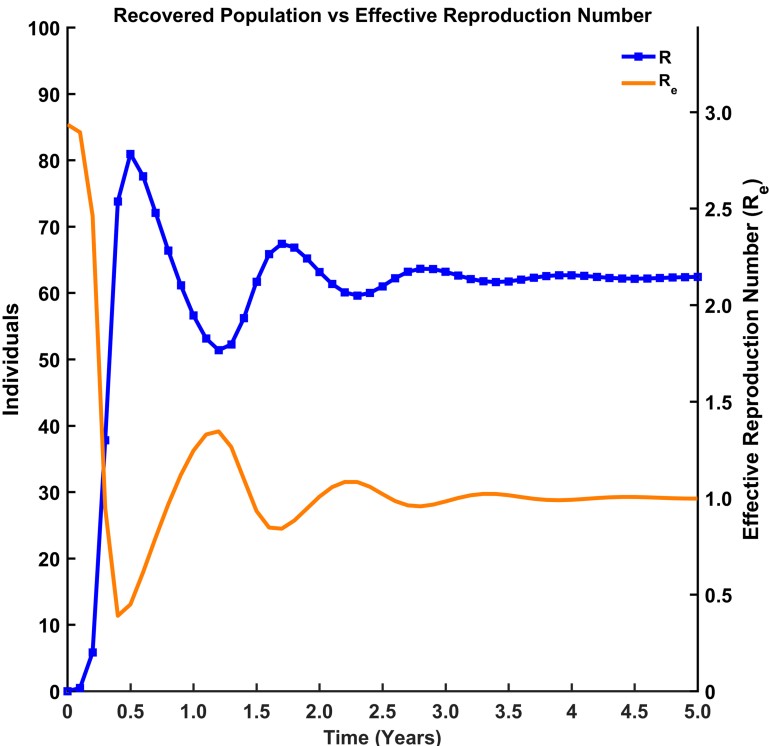

**Fig 18. Recovered population vs effective reproduction number.** Comparison between the recovered population ($R$) and $\mathcal{R}_e$.

of susceptible is lowest for the increasing gamma value. The disease spreads at the lowest rate if the recovery rate becomes high.

## Box plot analysis

A boxplot (box plot) is a graph that tells us how the data values are spread out. A succinct summary of the data distribution, including probable outliers and central tendencies, is given by this data visualization tool. In statistics, box plots are used to visually represent multiple parameters at a glance. A boxplot is often created to compare and contrast two or more groups. The box shows the range that contains the middle 50% of all data. As a result, the first quartile is at the bottom of the box, while the third quartile is at the top. In the boxplot, the solid line indicates the median. The T-shaped whiskers are the last point which indicates either maximum value or minimum value without outliers. Any points that are further away are considered outliers

Here boxplot analysis has been done to visualize the variability of the basic reproduction number. The box represents the interquartile range of the basic reproduction number ($\mathcal{R}_0$) and the typical value of $\mathcal{R}_0$ is the median. The range of $\mathcal{R}_0$ values can be known by the whiskers. In this section, we have carried out the box plot analysis of $\mathcal{R}_0$ as a function of two parameters [22].

Fig 28 shows the different values of $\mathcal{R}_0$ by varying parameters $\beta$ (transmission rate) and $\gamma$ (recovery rate). We notice that the relationship of $\mathcal{R}_0$ with the transmission rate ($\beta$). $\mathcal{R}_0$ increases with the increase of transmission rate, while this relationship is reciprocal for the case of the parameter $\gamma$. $\mathcal{R}_0$ increases when the recovery rate decreases. The interquartile

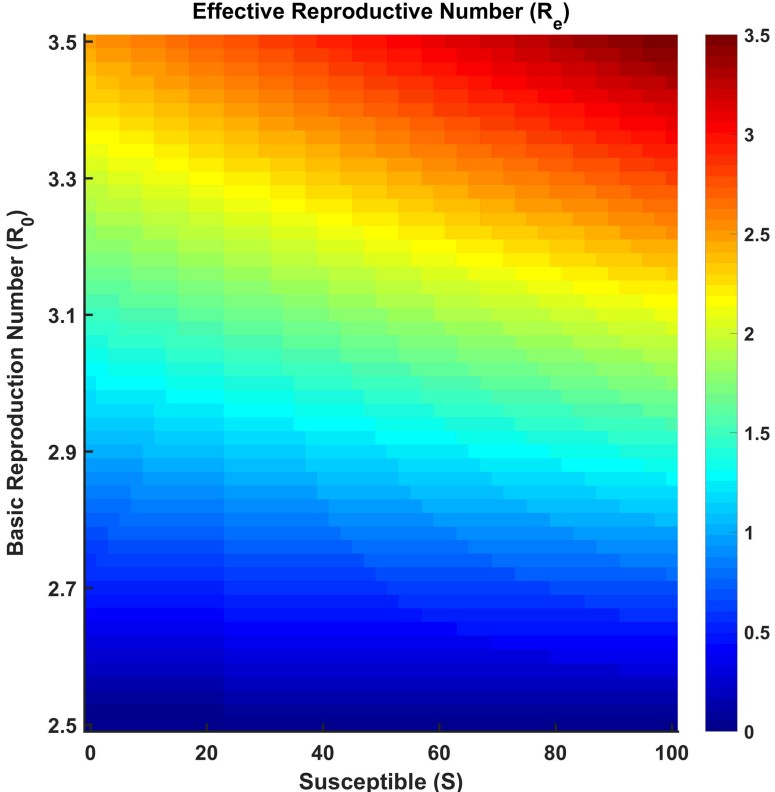

**Fig 19. Effective reproduction number ($R_e$).** The variation of the effective reproduction number ($\mathcal{R}_e$) as a function of the susceptible population ($S$) and basic reproduction number ($\mathcal{R}_0$).

range of $\mathcal{R}_0$ is 3.4887 to 2.387 and the median is 2.9378. Since there are no outliers, the range of $\mathcal{R}_0$ is [2.2034, 3.6723] which is still within 1.5 times the interquartile range. The maximum and minimum values of $\mathcal{R}_0$ are 3.6723 and 2.2034, respectively. The median value of $\mathcal{R}_0$ increases to 3.9165 by decreasing the recovery rate ($\gamma$) 25% from its exact value. In this case, the interquartile range and range of $\mathcal{R}_0$ become [3.1821, 4.6508] and [2.9374, 4.8956], respectively. A 25% rise in $\gamma$ leads to the median value of $\mathcal{R}_0$ falling to 2.3505 while the interquartile range and range become [1.9098, 2.7912] and [1.7629, 2.9381], respectively.

In Fig 29, $\mathcal{R}_0$ is depicted as a function of parameters $\beta$ and $\sigma$ using a box plot for better visualization. It indicates that the median or the typical value of $\mathcal{R}_0$ is 2.9378 when the range and the interquartile range are [2.2033, 3.6722] and [2.3869, 3.4886], respectively. If both parameters decrease to 25% from their actual value, the median becomes 2.9375 which means that the value of $\mathcal{R}_0$ decreases for the fall of transmission and latent rates. In this case, the interquartile range becomes [2.3867, 3.4883]. When both of these rates increase by 25%, the median of $\mathcal{R}_0$ is 2.9379, along with an interquartile range of 2.3871 to 3.4888 while the maximum and the minimum values are 3.6724 and 2.2034, respectively.

The box plot in Fig 30 displays the Basic Reproduction Number ($\mathcal{R}_0$) across three scenarios involving changes in the natural death rate ($\mu$) and latency rate ($\sigma$). The median value of $\mathcal{R}_0$ is 2.9378, while the interquartile range is [2.9374, 2.9382]. The median value of $\mathcal{R}_0$ is slightly lower, with a narrower range, indicating less variability in the values of $\mathcal{R}_0$ when the death rate decreases. But the interquartile range increases. When the death rate decreases by

**Fig 20. Effect of transmission rate ($\beta$) on $I(t)$ vs $S(t)$.** Phase plane of $I(t)$ vs. $S(t)$ compartments under varying transmission rate ($\beta$).

25%, the median value of $\mathcal{R}_0$ becomes 2.9375. The median $\mathcal{R}_0$ is still higher when the natural death rate decreases by 25%, but the range of $\mathcal{R}_0$ values appears to increase slightly and the median $\mathcal{R}_0$ is 2.9379. The basic reproduction number ($\mathcal{R}_0$) increases slightly as the natural death rate increases, with more variability when $\sigma$ is increased. Conversely, decreasing the death rate results in a lower and more stable $\mathcal{R}_0$. This suggests that changes in the natural death rate affect the spread of infection, with higher death rates potentially leading to a greater reproduction number, possibly because the disease progresses faster through the population.

### Contour plot analysis

A contour plot is a graphical technique used to represent three-dimensional data in two dimensions, often employed to show how a dependent variable behaves as two independent variables change. Contour plots are widely used in mathematics, physics, epidemiology, and other fields to display relationships between three quantities. A contour plot of $\mathcal{R}_0$ with respect to two parameters in epidemiology provides valuable insights into the spread and control of infectious diseases [23]. This helps us in this thesis work to understand the dynamics of disease transmission and make informed decisions regarding interventions and control measures [22,24]. It helps us to visualize the effect of two different independent parameters on the value of $\mathcal{R}_0$. In the color bar, each color represents a specific value of the dependent variable $\mathcal{R}_0$. The color intensity or shading within the contour plot indicates the relative magnitude of $\mathcal{R}_0$ at different points on the plot [23].

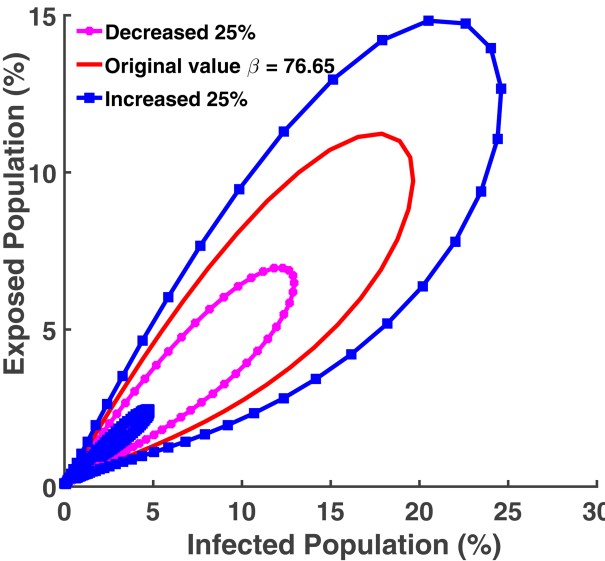

**Fig 21. Effect of transmission rate ($\beta$) on $I(t)$ vs $E(t)$.** Phase plane of $I(t)$ vs. $E(t)$ compartments under varying transmission rate ($\beta$).

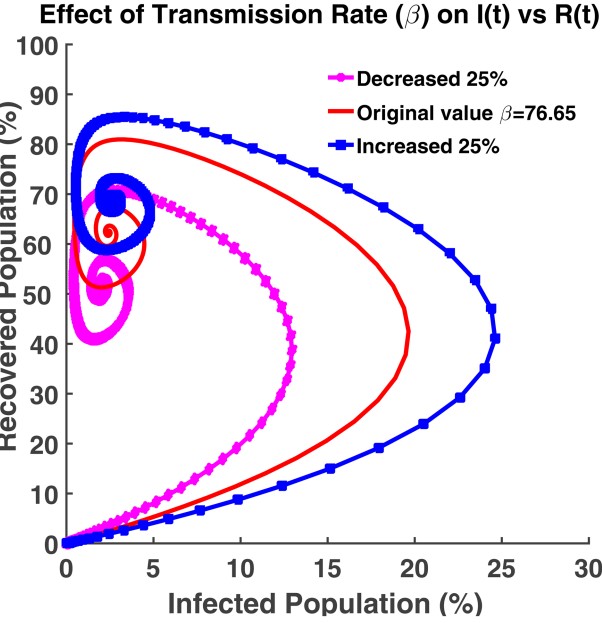

**Fig 22. Effect of transmission rate ($\beta$) on $I(t)$ vs $R(t)$.** Phase plane of $I(t)$ vs. $R(t)$ compartments under varying transmission rate ($\beta$).

This section includes a contour plot that maps the dependency of $\mathcal{R}_0$ on different input parameters. We also used a contour plot to visualize the effect of the transmission rate and recovery rate on the peak infected population. From Fig 31, we observe that the red color line

**Effect of Transmission Rate (β) on S(t) vs R(t)**

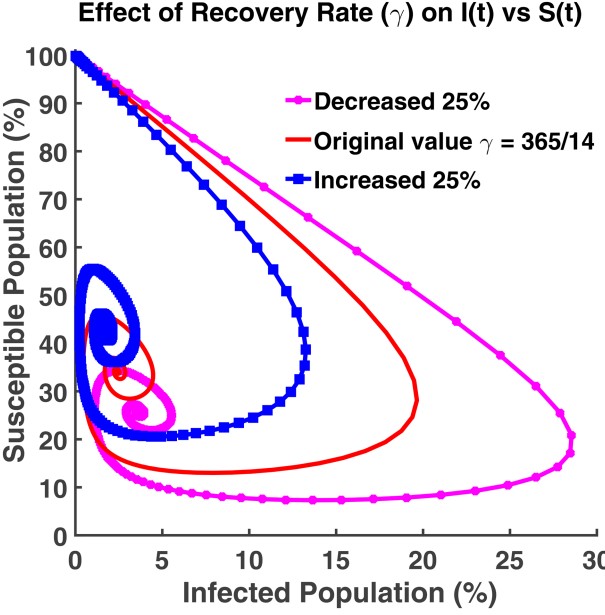

**Fig 23. Effect of transmission rate ($\beta$) on $S(t)$ vs $R(t)$.** Phase plane of $S(t)$ vs. $R(t)$ compartments under varying transmission rate ($\beta$).

**Effect of Recovery Rate (γ) on I(t) vs S(t)**

**Fig 24. Effect of recovery rate ($\gamma$) on $I(t)$ vs $S(t)$.** Phase plane of $I(t)$ vs. $S(t)$ compartments under varying recovery rate ($\gamma$).

indicates the highest percentage of the infected population which is obtained when the transmission rate ($\beta$) increases and the recovery rate ($\gamma$) decreases. Increasing $\gamma$ (moving upwards) reduces the peak infected population, as people recover faster, leaving fewer individuals in the infected state at any given time. Both the rate of transmission and recovery have an impact on

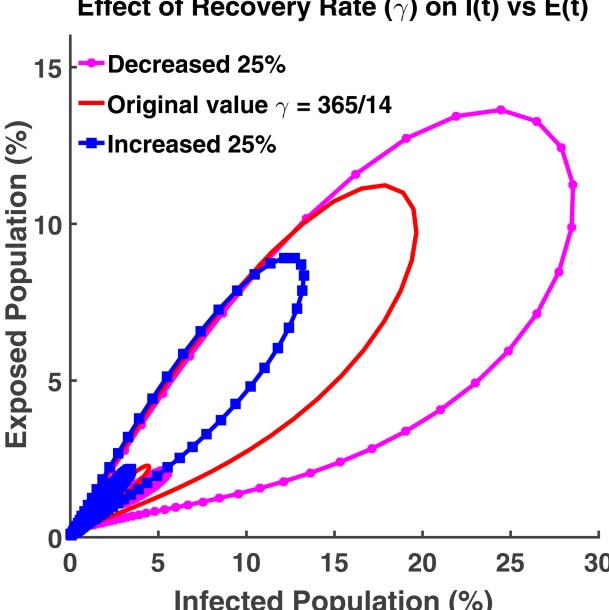

**Fig 25. Effect of recovery rate ($\gamma$) on $I(t)$ vs $E(t)$.** Phase plane of $I(t)$ vs. $E(t)$ compartments under varying recovery rate ($\gamma$).

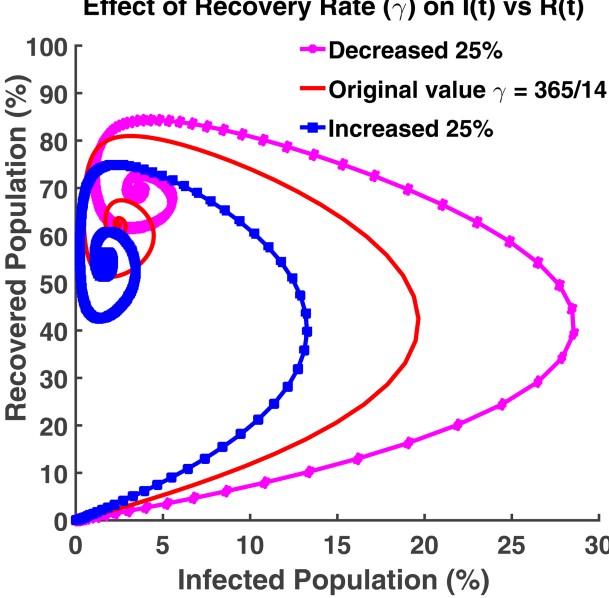

**Fig 26. Effect of recovery rate ($\gamma$) on $I(t)$ vs $R(t)$.** Phase plane of $I(t)$ vs. $R(t)$ compartments under varying recovery rate ($\gamma$).

the peak infected population. This suggests that controlling transmission (through measures like vaccination, masks, etc.), and improving recovery (through medical treatment) are key to managing outbreaks.

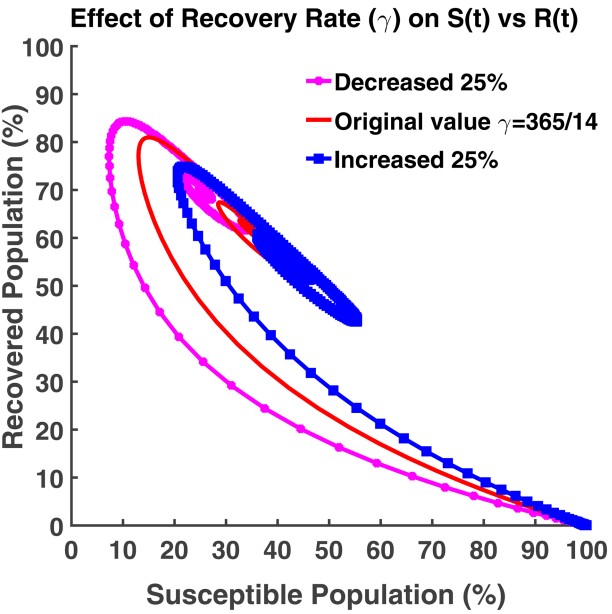

**Fig 27. Effect of recovery rate ($\gamma$) on $S(t)$ vs $R(t)$.** Phase plane of $S(t)$ vs. $R(t)$ compartments under varying recovery rate ($\gamma$).

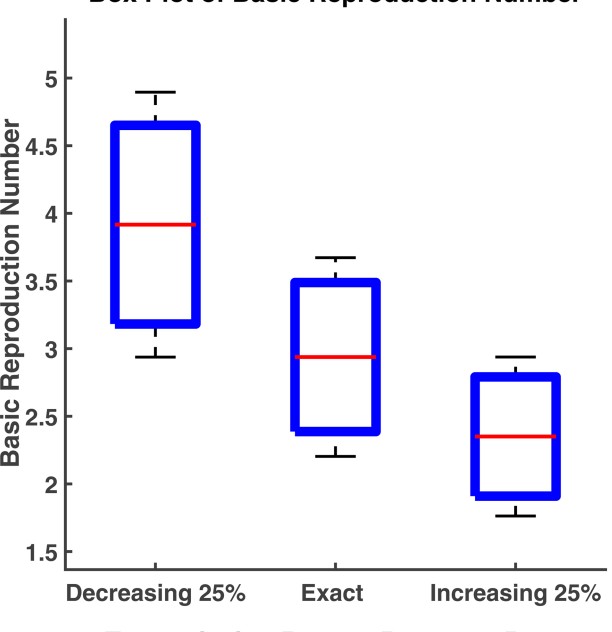

**Fig 28. Box plot of basic reproduction number.** Box plot of $\mathcal{R}_0$ vs. parameters $\beta$ and $\gamma$.

Fig 32 depicts how the basic reproduction number ($\mathcal{R}_0$) changes based on two key variables: the transmission rate ($\beta$) and the recovery rate ($\gamma$). The contour lines show the combined effect of transmission and recovery rates on $\mathcal{R}_0$. As the transmission rate increases and

**Box Plot of Basic Reproduction Number**

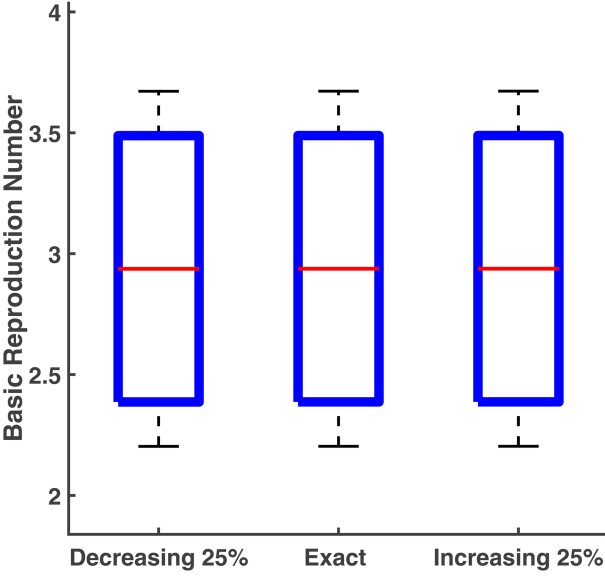

Fig 29. **Box plot of basic reproduction number.** Box plot of $\mathcal{R}_0$ vs. parameters $\beta$ and $\sigma$.

**Box Plot of Basic Reproduction Number**

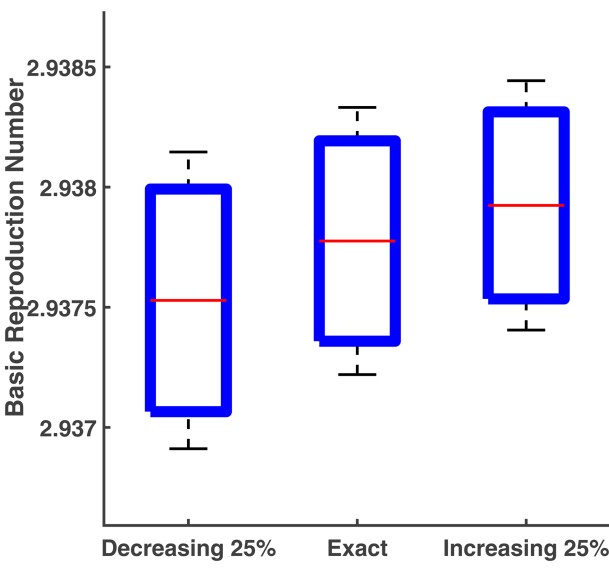

Fig 30. **Box plot of basic reproduction number.** Box plot of $\mathcal{R}_0$ vs. parameters $\mu$ and $\sigma$.

the recovery rate decreases, the values of $\mathcal{R}_0$ increase. As a result, the disease spreads at a faster rate. Managing these factors is crucial for controlling the spread of infection, as higher transmission and lower recovery will result in a larger $\mathcal{R}_0$.

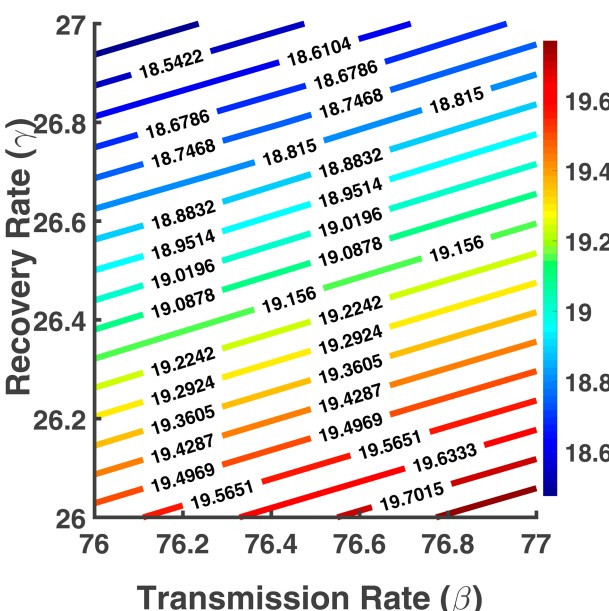

**Fig 31. Peak infected population in SEIRS model.** Contour plot analysis of $I(t)$ as a function of $\beta$ vs. $\gamma$.

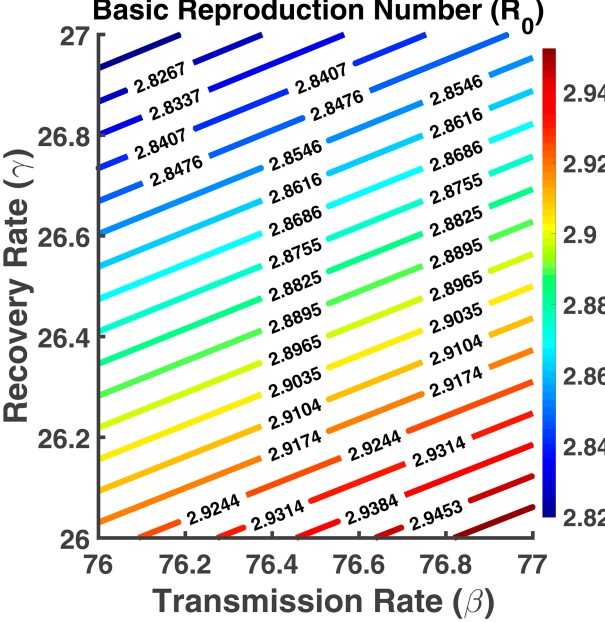

**Fig 32. Basic reproduction number ($R_0$).** Contour plot analysis of $\mathcal{R}_0$ as a function of $\beta$ vs. $\gamma$.

Fig 33 represents $\mathcal{R}_0$ as a function of two parameters $\beta$ and $\sigma$. From the contour plot, we see that the values of $\mathcal{R}_0$ increase if the latency rate increases. Higher values of $\beta$ and $\sigma$ lead to the highest $\mathcal{R}_0$ values. Fig 34 shows contour plots of $\mathcal{R}_0$ as function of parameters $\gamma$ and $\mu$. This indicates that the basic reproduction number increases when the recovery rate as well as

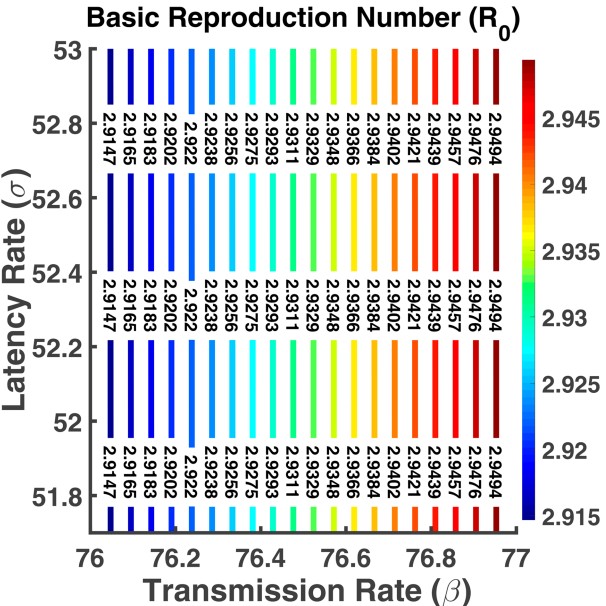

**Fig 33. Basic reproduction number ($R_0$).** Contour plot analysis of $\mathcal{R}_0$ as a function of $\beta$ vs. $\sigma$.

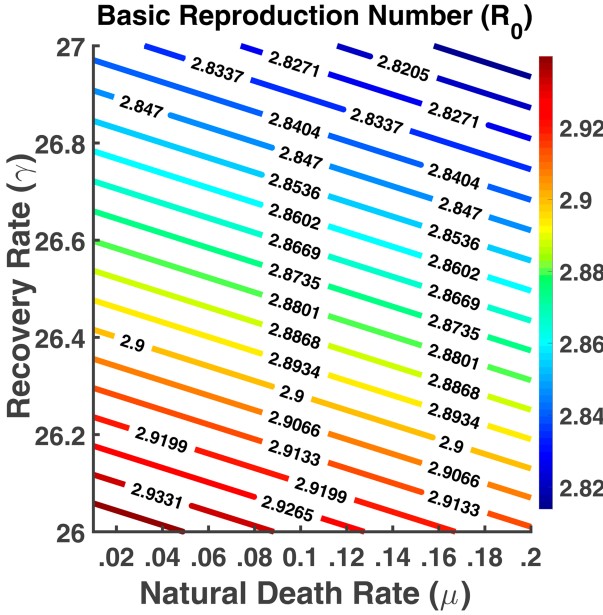

**Fig 34. Basic reproduction number ($R_0$).** Contour plot analysis of $\mathcal{R}_0$ as a function of $\mu$ vs. $\gamma$.

the natural death rate decreases. The spread of the disease depends on these parameters. $\mathcal{R}_0$ has a reverse relation with the progression rate of $\mu$ and $\gamma$. The value of $\mathcal{R}_0$ is changing rapidly between those parameters.

Fig 35 represents a contour plot showing the basic reproduction number as a function of latency rate ($\sigma$) and recovery rate ($\gamma$). Here the X-axis represents the latency rate and the Y-axis represents the recovery rate. We notice that the red color contour lines indicating the highest basic reproduction number are at the bottom while the recovery rate is the lowest and the latency rate is the highest. Here the control lines are parallel to the X-axis. The contour plot presented in Fig 36 illustrates the functional relationship between $\mathcal{R}_0$ and its associated parameters $\beta$ and $\omega$. Here $\mathcal{R}_0$ increases with the increase of the transmission and loss of immunity rates. A fall in immunity leads to the fastest spread of the disease.

The epidemiological interpretation suggests that increasing the recovery rates ($\gamma$) while decreasing the transmission rates ($\beta$), and the infection rate ($\sigma$) and gaining immunity can help control the disease spread. By doing so, the disease transmission is reduced, leading to fewer cases overall. This approach helps lower the burden on healthcare systems and the community. Faster recovery and lower transmission rates act as key factors in mitigating the epidemic's impact. Ultimately, these adjustments can significantly improve public health outcomes.

## Heatmap analysis

A heat map is a data visualization tool that uses colors to represent different values within a matrix. Heat maps typically use color gradients (e.g., from blue to red, or from light to dark) to represent numerical values or intensities. A color bar or legend will usually explain the meaning of the colors, such as blue for low values and red for high values. Heat maps often reveal correlations between variables.

The heat map displayed in Fig 37 represents the peak infected population of the model (1) in relation to the transmission rate ($\beta$) and the recovery rate ($\gamma$). The upper region represented by blue color indicates a smaller number of infected whereas the lower region

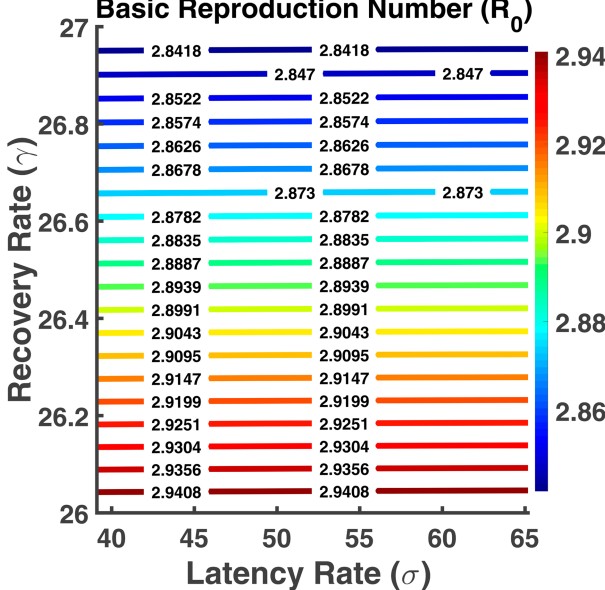

**Fig 35. Basic reproduction number ($R_0$).** Contour plot analysis of $\mathcal{R}_0$ as a function of $\sigma$ vs. $\gamma$.

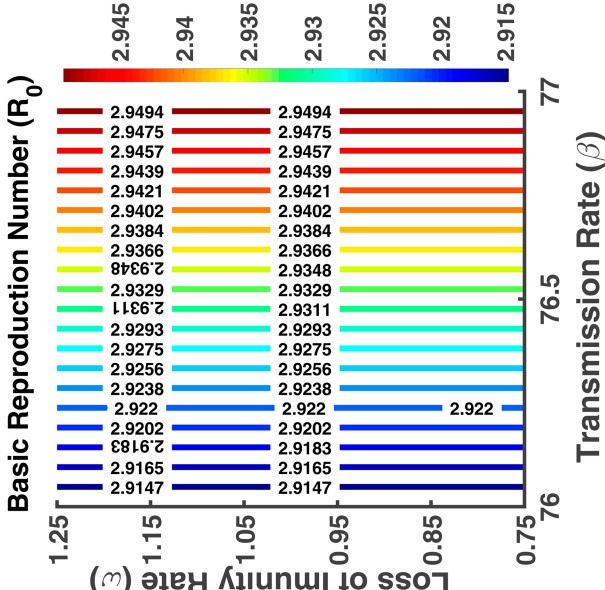

**Fig 36. Basic reproduction number ($R_0$).** Contour plot analysis of $\mathcal{R}_0$ as a function of $\beta$ vs. $\omega$.

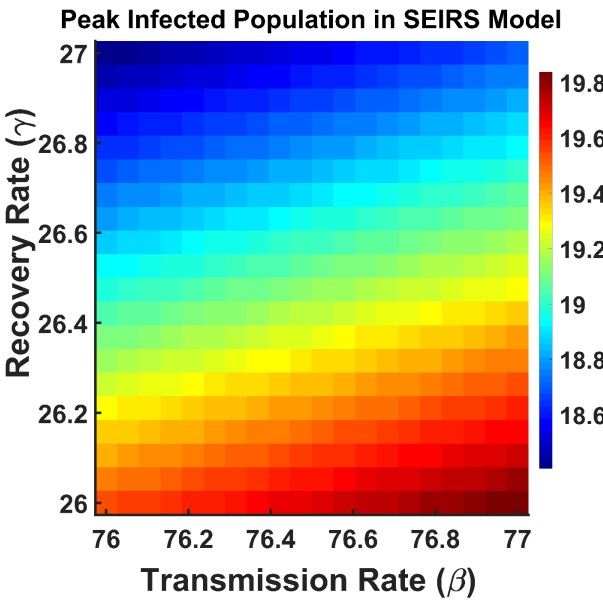

**Fig 37. Peak infected population in SEIRS model.** Heatmaps of $I(t)$ vs. parameters $\beta$ and $\gamma$.

represented by red color indicates a larger number of infected. This suggests that the number of affected populations rises when the transmission rate rises and the recovery rate falls.

The heat map provided in Fig 38 visualizes the basic reproduction number ($\mathcal{R}_0$) about the recovery rate ($\gamma$) and the transmission rate ($\beta$). The lower region of the heat map presented by red color indicates higher $\mathcal{R}_0$. This suggests that the higher transmission rate and the lower recovery rate increase the spread of the infection. The upper region of the map is blue,

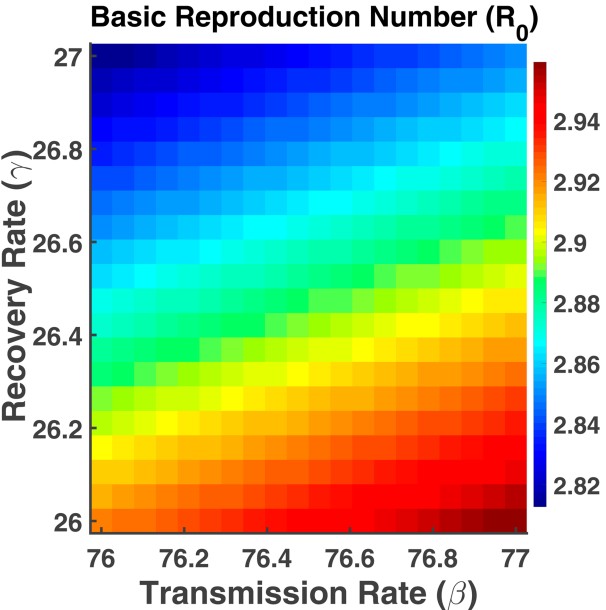

**Fig 38. Basic reproduction number ($R_0$).** Heatmaps of $\mathcal{R}_0$ vs. parameters $\mu$ and $\sigma$.

signifying a lower basic reproduction number. In this area, the spread of the infection is limited. There is a gradual shift from blue to red as we move from top-left to bottom-right. This indicates that increasing the transmission rate ($\beta$) or reducing the recovery rate ($\gamma$) leads to a higher basic reproduction number ($\mathcal{R}_0$), which correlates with a higher potential for disease spread. If the goal is to control the spread of infection by reducing $\mathcal{R}_0$, the ideal strategy would involve either reducing the transmission rate by interventions like social distancing or vaccination or increasing the recovery rate through effective treatment or boosting immunity.

Fig 39, representing the basic reproduction number as a function of the parameter $\sigma$ and $\mu$. The blue region indicates a lower basic reproduction number which occurs when the natural death rate is elevated. The red region represents a higher reproduction number which is obtained when the latency rate is very high and the natural death rate is very low. From the heatmap, it is clear that $\mathcal{R}_0$ is most influenced by changes in the natural death rate ($\mu$). Small changes in $\mu$ have a more pronounced effect on the reproduction number. The overall picture suggests that reducing natural death rates (through health interventions, for example) might sustain higher transmission rates while increasing the latency rate could also raise $\mathcal{R}_0$ but to a lesser degree.

## Influenza case study: California and North Carolina

The least squares approach is the most effective strategy to fit the suggested model with actual reported data [25]. For optimal curve fitting, the sum of squares of the vertical distances between the collected data and the data projected by the model should be as little as feasible. The formula of the sum of the squares error can be considered as,

$$g(\phi, n) = \sum_{i=1}^{n} \left( Y_i - I(t_i) \right)^2,$$

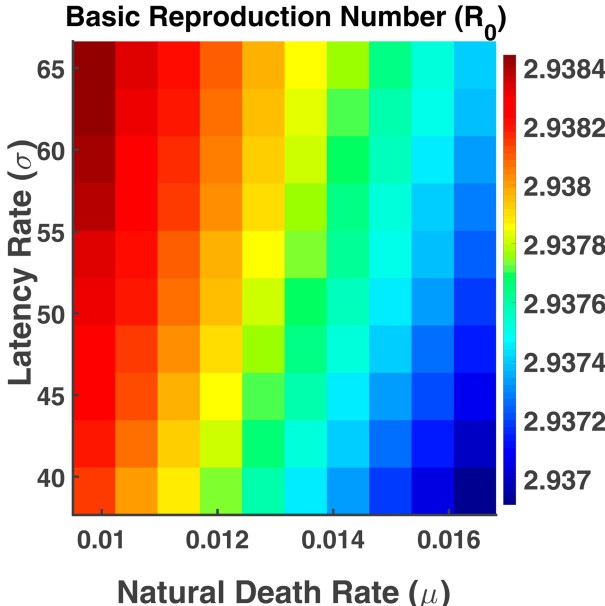

**Fig 39. Basic reproduction number ($R_0$).** Heatmaps of $\mathcal{R}_0$ vs. parameters $\sigma$ and $\gamma$.

where $\phi$ is the set of all model parameters, n represents the number of total observations, $I(t_i)$ indicates the number of models predicted infected population at the $i^{th}$ observation, and the $i^{th}$ observation, $Y_i$ represents the number of infected individuals observed in the population.

To validate the model, we have used actual cases of Influenza infection in two states of the USA namely California (see, S1 Data) and North Carolina (S2 Data) from 1st October 2023 to 21st September 2024 (total 356 days). We have collected this data from the CDC flu view web portal [18]. Our main goal is to estimate the parameters for forecasting the infection scenarios of these two states. In both figures, we show real data using a bar diagram and predicted data by line. Estimating these values analytically is very difficult. That's why we use the Matlab minimization command 'fminsearch' to estimate the parameter (minimize the residual). Considering $(S_0, E_0, I_0, R_0) = (98.14, 0.93, 0.93, 0)$ as the initial condition, parameters have been calculated for the case of California. Here initial values are taken in terms of percentage (%). Table 4 shows the estimated parameters for California.

Fig 40 portrays that the infection rate was at its peak on 23rd December 2023, which was approximately 18.32%. From 1st October 2023 to 23rd December 2023, the infection rate increased gradually, and then it started to decrease. It also illustrates that the number of infected people will begin to increase again after 6th July 2024. The basic reproduction number for these parameters is $\mathcal{R}_0 = 2.0097$ which is greater than unity. It suggests that the

**Table 4. Estimated values of the model parameters for California.**

| Parameter | Value | unit |
|---|---|---|
| $\beta$ | 0.939956 | $week^{-1}$ |
| $\sigma$ | 1.854173 | $week^{-1}$ |
| $\gamma$ | 0.451912 | $week^{-1}$ |
| $\mu$ | 0.012630 | $week^{-1}$ |
| $\omega$ | 0.024693 | $week^{-1}$ |

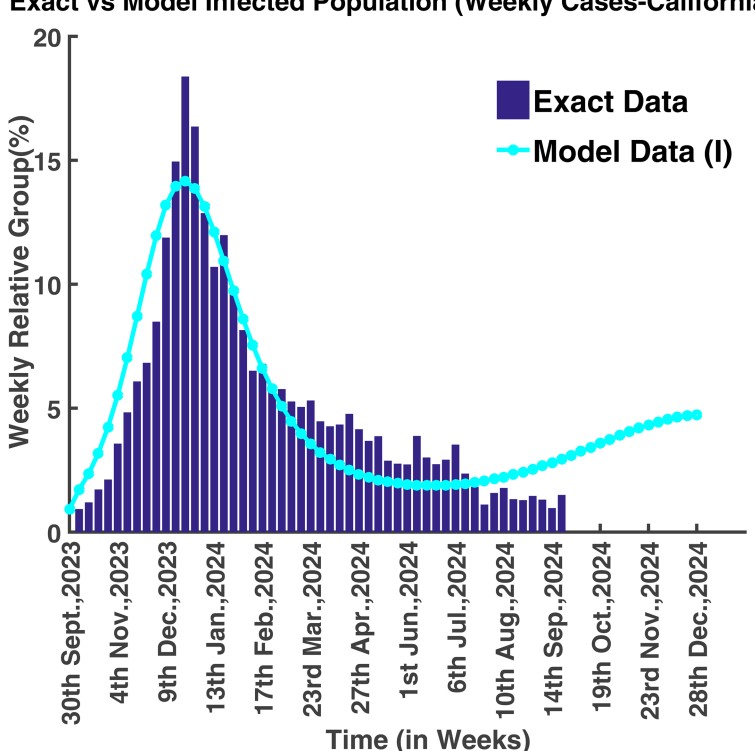

**Fig 40. Exact vs model infected population (weekly cases-California).** Model fit to actual data (California).

endemic equilibrium point is locally asymptotically stable which means that the disease will never die out, it will persist in nature.

Based on the initial condition of $(S_0, E_0, I_0, R_0) = (98.76, 0.62, 0.62, 0)$, parameters have been computed for the North Carolina scenario where these initial values are taken in terms of percentage (%). The estimated parameters for North Carolina are represented in Table 5.

Fig 41 predicted the long-term dynamical behavior of Influenza infection in California. We can observe that the number of infected population increased after 6th July 2024 and this will be at its peak on 18th January 2025. After 8th November 2025, the disease will remain under control and it will never die out.

Fig 42 illustrates that most of the people became infected in the 12th week (from 17th December 2023 to 23rd December 2023) which was approximately 21.14% of the total population. After 23rd December 2023, the transmission rate decreased. It continued up to 14th

**Table 5. Predicted parameter values for the model in North Carolina.**

| Parameter | Value | unit |
|---|---|---|
| $\beta$ | 1.338035 | $week^{-1}$ |
| $\sigma$ | 0.882401 | $week^{-1}$ |
| $\gamma$ | 0.491771 | $week^{-1}$ |
| $\mu$ | 0.009361 | $week^{-1}$ |
| $\omega$ | 0.010660 | $week^{-1}$ |

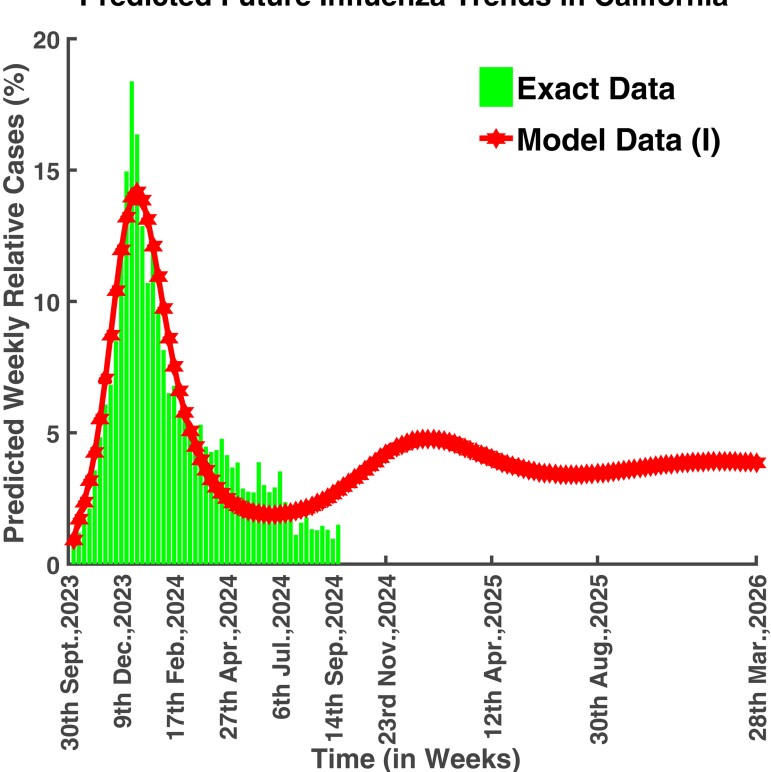

**Fig 41. Predicted future influenza trends in California.** Forecasted long-term dynamics of infected population (California).

September 2024. The epidemic was almost in control from 18th May 2024 to 14th September 2024. The number of infected people started to increase after 14th September 2024. The basic reproduction number ($\mathcal{R}_0$) for the scenario is also greater than one and this value is $\mathcal{R}_0$ = 2.642.

Fig 43 predicted the long-term dynamical behavior of Influenza infection in North Carolina. We can see that from September 14, 2024, the number of afflicted people rose, reaching its peak on April 12, 2025. After 28th March 2026, the disease will remain under control and it will persist in nature.

## Continuous-time Markov chains (CTMC)

Continuous-Time Markov Chains (CTMCs) are stochastic processes where transitions between states occur at continuous time intervals and depend only on the current state, not past states [26]. In epidemiology, CTMCs are used to model SEIRS dynamics probabilistically, capturing random transitions between susceptible, exposed, infectious, and recovered states. This is particularly valuable for modeling disease spread in small populations, where random fluctuations significantly affect outcomes [27,28]. If the basic reproduction number ($\mathcal{R}_0 < 1$), the CTMC-SEIRS process is ergodic, meaning it will eventually return to a disease-free equilibrium with probability 1 [29,30].

**Exact vs Model Infected Population (Weekly Cases-North Carolina)**

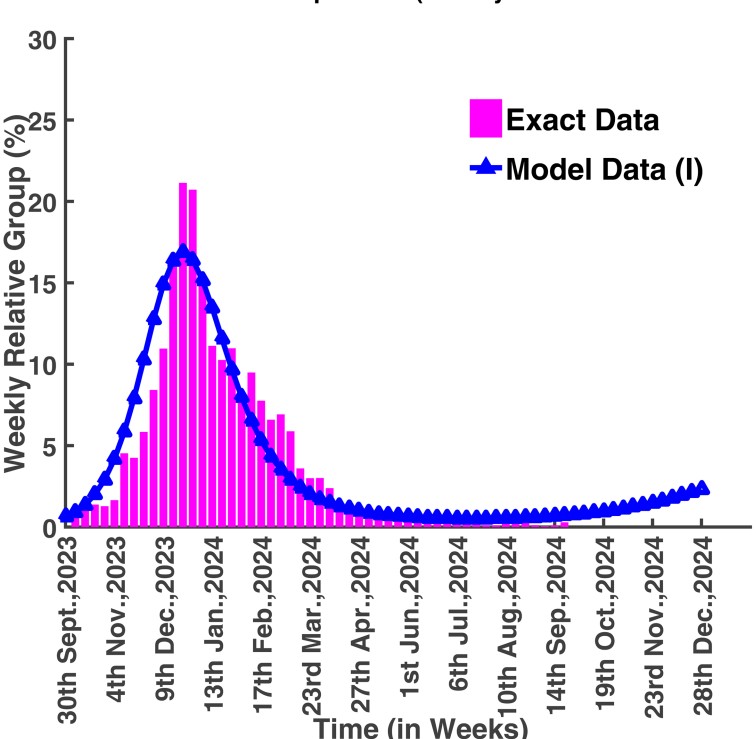

**Fig 42. Exact vs model infected population (weekly cases-North Carolina).** Model fit to actual data (North Carolina).

In continuous-time Markov chain (CTMC) models, particularly in epidemiological contexts such as the SEIRS (Susceptible-Exposed-Infectious-Recovered-Susceptible) model, transition probabilities play a fundamental role in defining the dynamics between different compartments. Transition probabilities dictate the likelihood of individuals moving between states (such as from susceptible to exposed, or exposed to infectious) over a small time interval, $\Delta t$. These probabilities are essential in understanding how diseases spread through populations and how recovery, infection, and immunity loss are governed.

Each transition in Table 6 is associated with a specific probability that governs the likelihood of movement between compartments. Below is a breakdown of the key transitions and their associated probabilities:

The natural population dynamics are modeled by the birth and death rates. A new birth adds an individual to the susceptible group ($\mu N$), while the death rate ($\mu S$) removes an individual from this group. The transition from the susceptible to the exposed class occurs when a susceptible individual contracts the disease. The probability of this transition is driven by the force of infection, expressed as $\beta \frac{SI}{N}$, where $\beta$ is the transmission rate, $S$ is the number of susceptibles, $I$ is the number of infectious individuals, and $N$ is the total population. This term models the interaction between susceptible and infectious individuals, leading to exposure.

Exposed individuals eventually become infectious after a latent period. The rate at which exposed individuals transition to the infectious class is governed by $\sigma E$, where $\sigma$ is the rate at which exposed individuals become infectious, and $E$ is the number of exposed individuals. Infectious individuals either recover or die. The probability of recovery is $\gamma I$, where $\gamma$

## Predicted Future Influenza Trends in North Carolina

**Fig 43. Predicted future influenza trends in North Carolina.** Forecasted long-term dynamics of infected population (North Carolina).

**Table 6. Transition probabilities for the CTMC-SEIRS model.**

| Event (*i*) | Description | ($\Delta S, \Delta E, \Delta I, \Delta R$) | Probability ($p_i$) |
|---|---|---|---|
| 1 | Birth of a Susceptible individual | $(+1, 0, 0, 0)$ | $\mu N$ |
| 2 | Death of a Susceptible individual | $(-1, 0, 0, 0)$ | $\mu S$ |
| 3 | Susceptible becomes Exposed | $(-1, +1, 0, 0)$ | $\beta \frac{SI}{N}$ |
| 4 | Death of an Exposed individual | $(0, -1, 0, 0)$ | $\mu E$ |
| 5 | Exposed becomes Infectious | $(0, -1, +1, 0)$ | $\sigma E$ |
| 6 | Death of an Infectious individual | $(0, 0, -1, 0)$ | $\mu I$ |
| 7 | Infectious individual recovers | $(0, 0, -1, +1)$ | $\gamma I$ |
| 8 | Death of a Recovered individual | $(0, 0, 0, -1)$ | $\mu R$ |
| 9 | Loss of immunity | $(+1, 0, 0, -1)$ | $wR$ |
| 10 | No change | $(0, 0, 0, 0)$ | $1 - \sum p_i \Delta t$ |

is the recovery rate, and *I* is the number of infectious individuals. The death rate for infectious individuals is denoted as $\mu I$. Recovered individuals may lose immunity and become susceptible again. This is captured by the transition probability $wR$, where *w* is the rate at which immunity is lost, and *R* is the number of recovered individuals. In any small time interval $\Delta t$, there is also the possibility that no transition occurs. The probability of no change is given by $1 - \sum p_i \Delta t$, ensuring that the total probability across all events sums to 1.

Fig 44 illustrates that, at present, stochastic and deterministic dynamics coexist and evolve in tandem. While their trajectories do not align perfectly, they exhibit a consistent sequence, maintaining a discernible correlation.

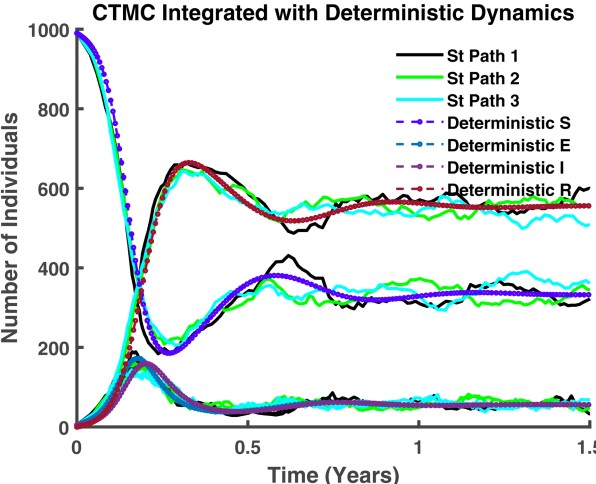

**Fig 44. CTMC integrated with deterministic dynamics.** Comparison of stochastic paths and deterministic trajectories in a CTMC model, depicting the evolution of susceptible, exposed, infected, and recovered compartments over 1.5 years.

In Figs 45-48, a more detailed observation reveals that, within individual compartments. Both dynamics adhere to similar pathways, demonstrating a high degree of parallelism in their behavior. Fig 45 illustrates the relationship between stochastic and deterministic dynamics for susceptible individuals. Similarly, Fig 46 represents this relationship for exposed individuals, while Fig 47 focuses on infectious individuals. Fig 48 depicts the corresponding dynamics for recovered individuals. Across all cases, a consistent pattern emerges, where the stochastic and deterministic approaches exhibit comparable trajectories, albeit with minor deviations.

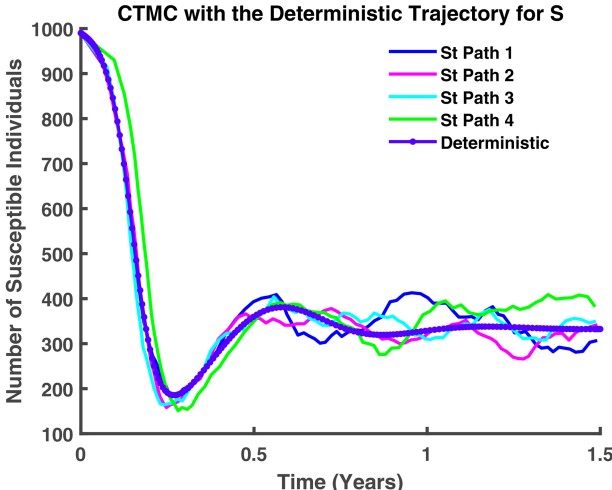

**Fig 45. CTMC with the deterministic trajectory for S.** Susceptible population (*S*) over time: stochastic (solid) vs deterministic (dashed).

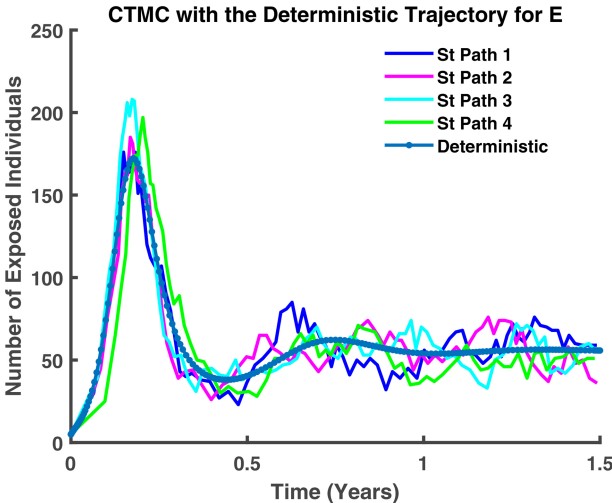

**Fig 46. CTMC with the deterministic trajectory for *E*.** Exposed population (*E*) over time: stochastic (solid) vs deterministic (dashed).

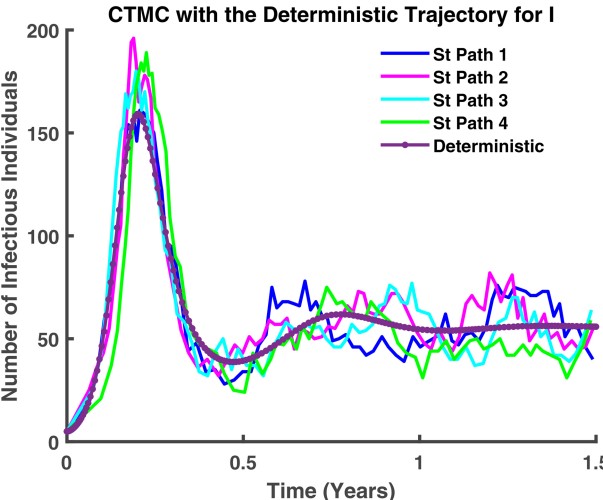

**Fig 47. CTMC with the deterministic trajectory for *I*.** Infected population (*I*) over time: stochastic (solid) vs deterministic (dashed).

## Discussion of computational works

We have plotted each compartment by varying different parameters such as recovery rate and transmission rate. These numbers enable us to see how slight variations in the rate of transmission or recovery impact the overall number of susceptible, exposed, infected, and recovered individuals. We found that while the susceptible, infected, and exposed population grows as the transmission rate increases, the recovered population increases as the recovery rate increases. Additionally, we have demonstrated the relationship between the basic reproduction number and changes in transmission and recovery rates. We have discovered that the basic reproduction number rises linearly with an increase in transmission rate and falls with

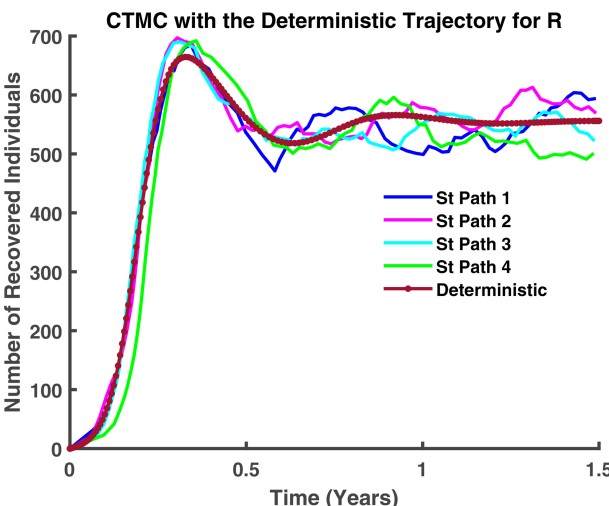

**Fig 48. CTMC with the deterministic trajectory for R.** Recovered population (*R*) over time: stochastic (solid) vs deterministic (dashed).

an increase in recovery rate. Phase plane analysis helps us to visualize how populations interact dynamically. These spiral-shaped curves suggest a potential for recurrent outbreaks, but in the long run, the disease will likely remain under control. The boxplot analysis provides us with the range of the basic reproduction number that determines the maximum or minimum number of exposed population that an infected individual infects at the initial stage of an outbreak. We may infer from the examination of contour plots and heat maps that a lower transmission rate and a quicker recovery rate are important components in reducing the impact of the epidemic. To forecast the long-term dynamics of influenza infections in North Carolina and California, we have estimated parameters using the least squares method. To determine the most sensitive parameter, a sensitivity analysis has been carried out using the Monte Carlo technique. We can compare the stochastic and deterministic situations with the aid of the CTMC approach.

## Conclusion

In conclusion, the SEIRS model offers a comprehensive framework for understanding the dynamics of influenza epidemics. We began by calculating the basic reproduction number, disease-free equilibrium, and endemic equilibrium and validated the model by checking the positivity and boundedness of the solutions. Our analysis revealed that the disease-free equilibrium is locally asymptotically stable if $\mathcal{R}_0 < 1$, and unstable otherwise. Conversely, the endemic equilibrium becomes stable when the basic reproduction number exceeds 1 (i.e., $\mathcal{R}_0 > 1$). We conducted a sensitivity analysis of each model parameter using PRCC values and corresponding p-values via the LHS method, identifying the death rate ($\mu$), transmission rate ($\beta$), and recovery rate ($\gamma$) as the most sensitive parameters. Graphical representations demonstrated that the transmission rate and recovery rate are particularly influential. Visual tools such as boxplots, contour plots, heat maps, and phase planes illustrated regions with the highest disease spread. The disease propagates most rapidly when the transmission rate ($\beta$) and latency rate ($\sigma$) increase while the recovery rate ($\gamma$) decreases. Moreover, graphically, we have shown the decline of the effective reproduction number in the SEIRS model over time, driven by reduced susceptibility and increased recovery, ultimately leading to epidemic control and

potential recurring outbreaks due to waning immunity. Stochastic behavior was examined using the CTMC method, showing alignment with the deterministic model. Parameter estimation through the least squares approach, using influenza data from California and North Carolina, confirmed the model's validity. The best-fitted curve indicated that our model is accurate. Since the basic reproduction number exceeds 1 in both cases, eradicating the disease is unfeasible. Therefore, implementing control strategies is essential to manage outbreaks. Reducing the transmission rate and increasing the recovery rate can mitigate the spread. Measures such as isolation, wearing masks, and using hand sanitizer can lower the transmission rate, while vaccination and boosting immunity can enhance the recovery rate. In the current model, we assume a homogeneous population and do not account for factors such as age, demographic and spatial variations, mobility, migration, vaccination, quarantine measures, or environmental conditions. Future work could address these limitations by incorporating heterogeneous populations, mobility patterns, and more complex interactions, thereby enhancing the model's realism and predictive accuracy.

## Supporting information

**S1 Appendix. Demonstrations of the basic reproduction number and fixed points.**
(PDF)

**S2 Appendix. Analytical stability and sensitivity study.**
(PDF)

**S1 Data. California Data File (CSV).**
(CSV)

**S2 Data. North Carolina Data File (CSV).**
(CSV)

## Author contributions

**Conceptualization:** Asma Akter Akhi, Rabeya Akther Diba.

**Data curation:** Rabeya Akther Diba, Mohammed Abid Anwar.

**Formal analysis:** Asma Akter Akhi, Rabeya Akther Diba.

**Funding acquisition:** Md. Kamrujjaman.

**Investigation:** Asma Akter Akhi, Mohammed Abid Anwar, Tarik Mahmud Akash, Md. Kamrujjaman.

**Methodology:** Asma Akter Akhi, Rabeya Akther Diba, Mohammed Abid Anwar, Tarik Mahmud Akash.

**Project administration:** Mohammed Abid Anwar.

**Software:** Asma Akter Akhi, Rabeya Akther Diba, Mohammed Abid Anwar, Tarik Mahmud Akash.

**Supervision:** Md. Kamrujjaman.

**Validation:** Tarik Mahmud Akash, Md. Kamrujjaman.

**Visualization:** Tarik Mahmud Akash.

**Writing – original draft:** Asma Akter Akhi.

**Writing – review & editing:** Md. Kamrujjaman.

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
