## [Decision Letter · Decision Letter 0]

2 May 2025

PGPH-D-25-00597

State-by-State Influenza Outbreaks and Oversee: A Markov Chain Study of California and North Carolina, USA

Dear Dr. Kamrujjaman,

Thank you for submitting your manuscript to PLOS Global Public Health. After careful consideration, we feel that it has merit but does not fully meet PLOS Global Public Health’s publication criteria as it currently stands. Therefore, we invite you to submit a revised version of the manuscript that addresses the points raised during the review process.

We look forward to receiving your revised manuscript.

Kind regards,

Arinjita Bhattacharyya

Guest Editor

Journal Requirements:

1. Please note that PLOS Global Public Health has specific guidelines on code sharing for submissions in which author-generated code underpins the findings in the manuscript. In these cases, all author-generated code must be made available without restrictions upon publication of the work. Please review our guidelines at https://journals.plos.org/globalpublichealth/s/materials-and-software-sharing#loc-sharing-code and ensure that your code is shared in a way that follows best practice and facilitates reproducibility and reuse. 2. We ask that a manuscript source file is provided at Revision. Please upload your manuscript file as a .doc, .docx, .rtf or .tex. 3. Your manuscript is missing the following sections: Method. Please ensure these are present, and in the correct order, and that any references to subheadings in your main text are correct. An outline of the required sections can be consulted in our submission guidelines here: https://journals.plos.org/globalpublichealth/s/submission-guidelines#loc-parts-of-a-submission

Additional Editor Comments (if provided):

Reviewers' comments:

Reviewer's Responses to Questions

**Comments to the Author**

1. Does this manuscript meet PLOS Global Public Health’s publication criteria? Is the manuscript technically sound, and do the data support the conclusions? The manuscript must describe methodologically and ethically rigorous research with conclusions that are appropriately drawn based on the data presented.

Reviewer #1: Yes

Reviewer #2: Yes

2. Has the statistical analysis been performed appropriately and rigorously?

Reviewer #1: Yes

Reviewer #2: Yes

3. Have the authors made all data underlying the findings in their manuscript fully available (please refer to the Data Availability Statement at the start of the manuscript PDF file)?

Reviewer #1: Yes

Reviewer #2: Yes

4. Is the manuscript presented in an intelligible fashion and written in standard English?

Reviewer #1: Yes

Reviewer #2: Yes

5. Review Comments to the Author

Reviewer #1: In this study, the authors present a robust SEIRS mathematical model to analyze influenza transmission dynamics, using data from influenza cases in California and North Carolina between October 1, 2023, and September 24, 2024. This comprehensive study combines both deterministic and stochastic frameworks, making it relevant for public health planning.

A few minor comments are provided below:

For clarity, it would be beneficial if the authors explicitly define all parameters and introduce a schematic of the transmission dynamics in Section 2 of the mathematical models. Alternatively, they could refer to Table 1 and Figure 2 in the same section.

Several additional factors can affect transmission, including individuals’ age, spatial variations of infectious agents, co-infections, and mobility. Does this model also take these into account? Please discuss the limitations of the model in detail.

Reviewer #2: Authors have explained the transmission of influenza with the SEIR mathematical model perfectly. The equations, plots and their findings establish very clearly the spread of influenza and they validate it with actual case studies of North Carolina and California. The article provides full clarity on each of their parameters, equations and models to explain the transmission dynamics of Influenza.

Minor comment:

I would be interested in seeing a summary figure like what we usually see in epidemiology papers giving counts of individuals say 100 individuals for example who are susceptible (S), then this 100 becomes what in each step of SEIR flow, i.e. at E, I and R. Also, if they can incorporate how basic reproduction number changes at each step of S, E, I and R and finally it becomes lowest as recovery rate (gamma) increases into this figure, that would be great if possible. This figure should be able to explain the big picture of the whole paper like a graphical abstract which can go into Conclusions maybe.

6. PLOS authors have the option to publish the peer review history of their article (what does this mean?). If published, this will include your full peer review and any attached files.

**Do you want your identity to be public for this peer review?** For information about this choice, including consent withdrawal, please see our Privacy Policy.

Reviewer #1: No

Reviewer #2: No

---

## [Decision Letter · Decision Letter 1]

11 Aug 2025

State-by-State Influenza Outbreaks and Oversee: A Markov Chain Study of California and North Carolina, USA

PGPH-D-25-00597R1

Dear Dr. Kamrujjaman,

We are pleased to inform you that your manuscript 'State-by-State Influenza Outbreaks and Oversee: A Markov Chain Study of California and North Carolina, USA' has been provisionally accepted for publication in PLOS Global Public Health.

Best regards,

Julia Robinson

Executive Editor

Reviewer Comments (if any, and for reference):

Reviewer's Responses to Questions

**Comments to the Author**

1. If the authors have adequately addressed your comments raised in a previous round of review and you feel that this manuscript is now acceptable for publication, you may indicate that here to bypass the “Comments to the Author” section, enter your conflict of interest statement in the “Confidential to Editor” section, and submit your "Accept" recommendation.

Reviewer #1: All comments have been addressed

Reviewer #2: All comments have been addressed

2. Does this manuscript meet PLOS Global Public Health’s publication criteria? Is the manuscript technically sound, and do the data support the conclusions? The manuscript must describe methodologically and ethically rigorous research with conclusions that are appropriately drawn based on the data presented.

Reviewer #1: Yes

Reviewer #2: Yes

3. Has the statistical analysis been performed appropriately and rigorously?

Reviewer #1: Yes

Reviewer #2: Yes

4. Have the authors made all data underlying the findings in their manuscript fully available (please refer to the Data Availability Statement at the start of the manuscript PDF file)?

Reviewer #1: Yes

Reviewer #2: Yes

5. Is the manuscript presented in an intelligible fashion and written in standard English?

Reviewer #1: Yes

Reviewer #2: Yes

6. Review Comments to the Author

Reviewer #1: (No Response)

Reviewer #2: (No Response)

7. PLOS authors have the option to publish the peer review history of their article (what does this mean?). If published, this will include your full peer review and any attached files.

**Do you want your identity to be public for this peer review?** For information about this choice, including consent withdrawal, please see our Privacy Policy.

Reviewer #1: No

Reviewer #2: No
